# Observations of a Magellanic Corona

Dhanesh Krishnarao[1,2,3✉], Andrew J. Fox[4], Elena D'Onghia[5], Bart P. Wakker[5], Frances H. Cashman[1], J. Christopher Howk[6], Scott Lucchini[7], David M. French[1] & Nicolas Lehner[6]

The Large Magellanic Cloud (LMC) and the Small Magellanic Cloud (SMC) are the closest massive satellite galaxies of the Milky Way. They are probably on their first passage on an infalling orbit towards our Galaxy[1] and trace the continuing dynamics of the Local Group[2]. Recent measurements of a high mass for the LMC ($M_{halo} \approx 10^{11.1-11.4}\,M_{\odot}$)[3–6] imply that the LMC should host a Magellanic Corona: a collisionally ionized, warm-hot gaseous halo at the virial temperature ($10^{5.3-5.5}$ K) initially extending out to the virial radius (100–130 kiloparsecs (kpc)). Such a corona would have shaped the formation of the Magellanic Stream[7], a tidal gas structure extending over 200° across the sky[2,8,9] that is bringing in metal-poor gas to the Milky Way[10]. Here we show evidence for this Magellanic Corona with a potential direct detection in highly ionized oxygen ($O^{+5}$) and indirectly by means of triply ionized carbon and silicon, seen in ultraviolet (UV) absorption towards background quasars. We find that the Magellanic Corona is part of a pervasive multiphase Magellanic circumgalactic medium (CGM) seen in many ionization states with a declining projected radial profile out to at least 35 kpc from the LMC and a total ionized CGM mass of $\log_{10}(M_{HII,CGM}/M_{\odot}) \approx 9.1 \pm 0.2$. The evidence for the Magellanic Corona is a crucial step forward in characterizing the Magellanic group and its nested evolution with the Local Group.

We use a sample of 28 Hubble Space Telescope (HST)/Cosmic Origins Spectrograph (COS) spectra of background UV-bright quasars within an angular separation of the LMC of 45°, corresponding to an impact parameter $\rho_{LMC} < 35$ kpc, spanning one-third of the initial virial radius of the LMC. Six of these sightlines also have archival Far Ultraviolet Spectroscopic Explorer (FUSE) spectra with high enough signal-to-noise (S/N) ratio to measure O VI absorption. All spectra have a S/N ratio >7 per resolution element. Our analysis shows pervasive low-ion and high-ion absorption in several phases centred around the LMC, both on the sky and in velocity, with radial velocities distinct from the Milky Way. The C IV absorption has a high covering fraction of 78% within 25 kpc, but in the range 25 kpc < $\rho_{LMC}$ < 35 kpc, the covering fraction is 30%. Figure 1 shows two maps of the Magellanic system superimposed on 21-cm H I emission maps of neutral hydrogen[11], with our sightline locations colour-coded by C IV column density and mean velocity. We limit our analysis of the high-ion absorption to components with the following properties: (1) high-ion column densities not explained by photoionization; (2) velocities $v_{LSR} > 150$ km s$^{-1}$ to select Magellanic gas and avoid Milky Way contamination[12–14]; (3) velocities not associated with known intermediate-velocity and high-velocity clouds[13,15].

Photoionization accounts for the low-ion absorbers (singly and doubly ionized species) with temperatures $\log_{10}(T_e/K) = 4.02^{+0.07}_{-0.04}$, densities $\log_{10}(n_e/cm^{-3}) = -1.4 \pm 0.3$ and line-of-sight cloud sizes $\log_{10}(L/kpc) = -1.5^{+0.6}_{-0.4}$. The high-ion absorbers have column densities too high to be explained by photoionization and, instead, are well explained by equilibrium or non-equilibrium (time-dependent) collisional ionization models[16]. The C IV/Si IV column-density ratio

primarily yields a temperature of $T \approx 10^{4.9}$ K, but solutions as low as $T \approx 10^{4.3}$ K are possible in non-equilibrium conditions for certain metallicities. However, the measured ratios of O VI/C IV and O VI/Si IV yield higher temperatures. Our ionization modelling and the component kinematics instead suggest that the O VI ions are tracing a distinct and hotter phase of gas near about $10^{5.5}$ K, in which O VI peaks in fractional abundance in collisional ionization[16].

We show the relation between the C IV, Si IV and O VI column densities in the Magellanic CGM and the LMC impact parameter in Fig. 2. A significant declining profile is observed for C IV and Si IV, indicating that the gas content decreases with radius, a characteristic signature of a diffuse CGM[17]. Sightlines at small impact parameters of $\rho_{LMC} < 7$ kpc tend to show a deficit of collisionally ionized high ions compared with sightlines at 7 kpc < $\rho_{LMC}$ < 12 kpc. These inner-CGM absorbers are more susceptible to photoionization and winds, owing to their close proximity to the LMC. When only considering the absorbers at $\rho_{LMC} > 7$ kpc, the significance of the anti-correlation between $N$(Si IV) and $\rho_{LMC}$ becomes stronger. A similar trend is seen with O VI, but is more uncertain owing to the small sample size.

For the approximately $10^4$ K low-ion CGM phase, the relation between the modelled ionized hydrogen column density ($N_{HII}$) and the impact parameter ($\rho_{LMC}$) is shown in Fig. 3a. The ionized hydrogen column densities for the approximately $10^{4.9}$K CGM phase traced by C IV and Si IV and derived from both an equilibrium and non-equilibrium collisional ionization model[16] are shown in Fig. 3b. Both the low-ion and high-ion gas show similar radial profiles and all but one sightline in which high-ion absorption is observed also show low-ion absorption.

[1]Space Telescope Science Institute, Baltimore, MD, USA. [2]William H. Miller III Department of Physics & Astronomy, Johns Hopkins University, Baltimore, MD, USA. [3]Department of Physics, Colorado College, Colorado Springs, CO, USA. [4]AURA for ESA, Space Telescope Science Institute, Baltimore, MD, USA. [5]Department of Astronomy, University of Wisconsin–Madison, Madison, WI, USA. [6]Department of Physics and Astronomy, University of Notre Dame, Notre Dame, IN, USA. [7]Department of Physics, University of Wisconsin–Madison, Madison, WI, USA. ✉e-mail: dkrishnarao@coloradocollege.edu

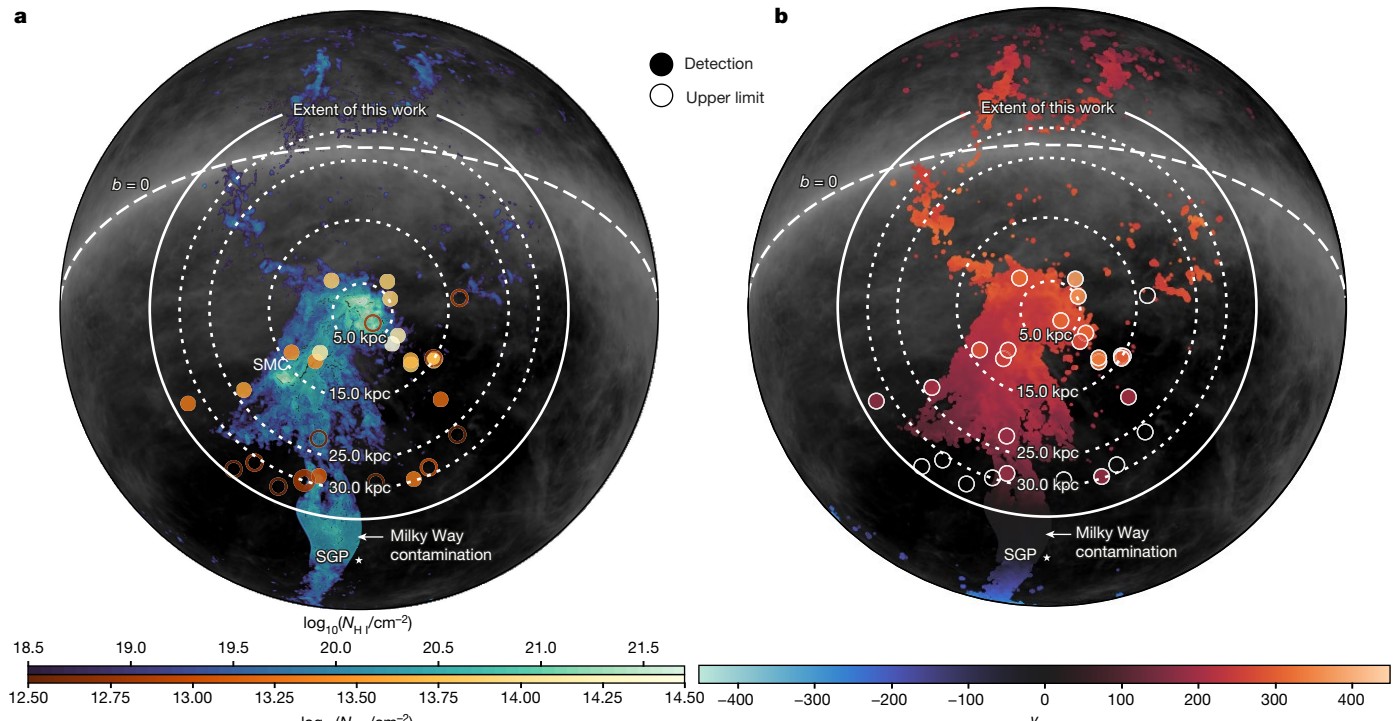

**Fig. 1 | The Magellanic system in an orthographic projection.** The maps are centred on the LMC and colour-coded by column-density (**a**) and velocity (**b**). Magellanic 21-cm H I emission is shown in a blue scale integrated in velocities to encompass the Magellanic system[9], with coloured symbols showing HST/COS sightlines colour-coded by C IV column densities. Upper limits are shown using open symbols. The grey background shows Galactic 21-cm H I emission from HI4PI[11] integrated over $-75 < v_{LSR} < +75$ km s$^{-1}$. Panel **b** shows the mean velocity of H I with our sightlines colour-coded by mean C IV absorption velocity. Dotted circles mark the LMC impact parameter and the South Galactic Pole (SGP) is marked with a white star.

A linear regression model is used to find the ionized hydrogen mass of these two phases within $\rho_{LMC} < 35$ kpc, using a measured metallicity of [Z/H] = −0.67 for the photoionized gas and an assumed metallicity of [Z/H] = −1 for the high-ion gas. We find a total ionized hydrogen mass in the approximately $10^4$ K phase of $\log_{10}(M_{H\,II,photo}/M_\odot) = 8.7^{+0.2}_{-0.1}$. For the approximately $10^{4.9}$ K phase, we find a similar total ionized hydrogen mass of $\log_{10}(M_{H\,II,Eq}/M_\odot) = 8.5 \pm 0.1$ from an equilibrium model, whereas an isochoric non-equilibrium model results in

$\log_{10}(M_{H\,II,Non\text{-}Eq}/M_\odot) = 8.6 \pm 0.1$. If this high-ion gas was instead at a lower temperature of about $10^{4.3}$ K, as is possible in the isochoric model, the mass would increase by an order of magnitude.

For the approximately $10^{5.5}$ K CGM phase traced by O VI, we again assume a metallicity of [Z/H] = −1 and derive an ionized hydrogen mass within the range of LMC impact parameters in which O VI is observed to be $\log_{10}(M_{H\,II}/M_\odot) \approx 8.3 \pm 0.1$ for the equilibrium and $\log_{10}(M_{H\,II}/M_\odot) \approx 8.5 \pm 0.1$ for the isochoric non-equilibrium models. As we do not expect any of our high-ion gas phases to maintain collisional ionization equilibrium, the non-equilibrium case is more likely. Combined across phases, we estimate a total ionized Magellanic CGM gas mass of $\log_{10}(M_{H\,II,CGM}/M_\odot) \approx 9.1 \pm 0.2$.

We consider three possible explanations for the C IV-bearing and Si IV-bearing gas around the LMC. It could exist in: (a) a diffuse Magellanic Corona at approximately $10^5$ K, (b) turbulent or conductive interfaces[18,19] between cool, low-ion gas clouds and a hotter diffuse Magellanic Corona at approximately $10^{5.5}$ K or (c) turbulent or conductive interfaces between cool, low-ion gas clouds and a hot (approximately $10^6$ K) gaseous Milky Way halo. Figure 4 shows a cartoon schematic of these three scenarios. When considering all our observations, we conclude that the C IV exists in the interfaces between cooler approximately $10^4$ K clouds and an approximately $10^{5.5}$ K Magellanic Corona (model b). This model explains the high-ion radial profile, because there are more cool clouds closer into the LMC, each with interfaces tracking the kinematics of low ions. It also explains the presence of O VI, which shows velocity offsets and directly traces the approximately $10^{5.5}$ K corona, but some of which may exist in interfaces. The linewidths are also consistent, with Si IV linewidths that are broader than the Si II linewidths, as expected with interfaces (though a difference is not confirmed at high significance between C IV and C II linewidths). The alternative models are less favoured because they either result in a thermally unstable corona with no explanation for O VI (model a) or

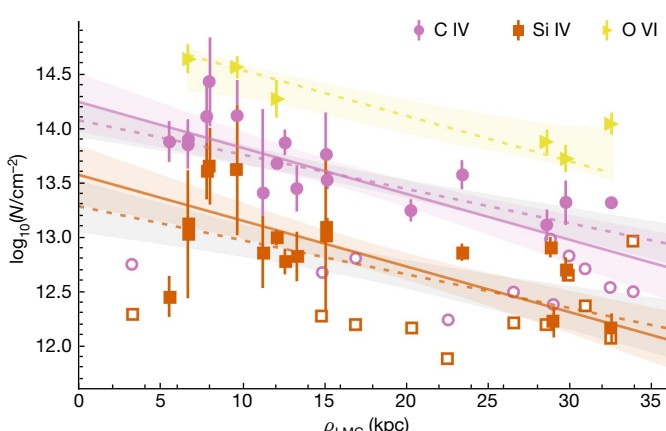

**Fig. 2 | Declining radial profile of high-ion absorption.** Sum total of O VI (yellow triangles), C IV (pink circles) and Si IV (orange squares) column density of Magellanic velocity absorbers that are not photoionized in each sightline as a function of LMC impact parameter, $\rho_{LMC}$, with open symbols marking upper limits. The best-fit line and 68% confidence intervals for all data (dotted lines) and only data at $\rho_{LMC} > 7$ kpc (solid lines) are also shown (see Methods). Error bars represent standard deviations.

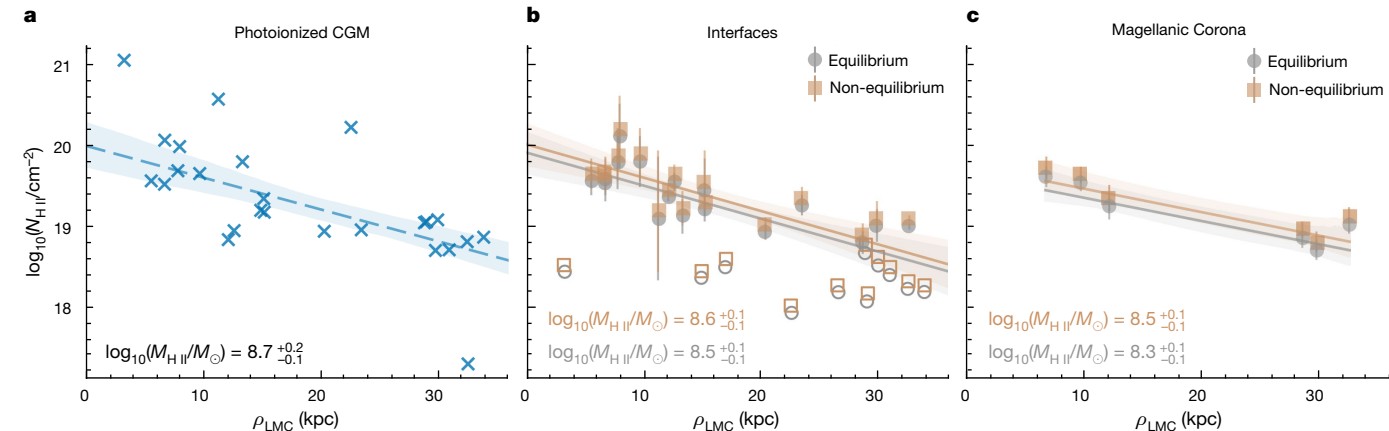

**Fig. 3 | Total ionized hydrogen profile from ionization models.** Ionized hydrogen column density as a function of LMC impact parameter, $\rho_{LMC}$, for each of three gas phases at approximately $10^4$ K (photoionized; low ions; **a**), approximately $10^{4.9}$ K (interfaces; C IV and Si IV; **b**) and approximately $10^{5.5}$ K (corona; O VI; **c**). The photoionized results are from Cloudy models (blue crosses). Collisional ionization equilibrium (grey circles) and non-equilibrium isochoric models (brown squares) are used for the approximately $10^{4.9}$ K and

approximately $10^{5.5}$ K gases based on C IV and O VI measurements, respectively[16]. Linear regression fits and 68% confidence intervals are shown in the same colour lines, with the roughly $10^{4.9}$ K fit made only to data at $\rho_{LMC} > 7$ kpc and the approximately $10^4$ K and approximately $10^{5.5}$ K fits made to all data. The slope of the approximately $10^{5.5}$ K fit is only considered within the bounds of our observations. Error bars represent standard deviations.

cannot explain the radial profile dependent on distance from the LMC, not distance from the Milky Way (model c).

Several sightlines in our sample have been studied as part of a survey of the Magellanic Stream[10], in which some high-ion absorption may be attributed to photoionization associated with a Seyfert flare in the Milky Way[20,21] or to a shock cascade[22]. However, the modelled Seyfert flare only affects gas within a relatively small ionization cone outside our sample. Gas associated with the Stream may contribute to the absorption measured in several other sightlines in our sample, but—overall—the tidally stripped Stream gas is less likely to be a principal contributor to the Magellanic CGM, as it cannot easily explain the radial profile seen in both low and high ions. The radial profile is also seen in sightlines

away from the Stream, further supporting that our observations are not biased by the tidally stripped gas. This radial profile seems truncated when compared with those seen in a previous survey of the CGM of 43 low-mass, $z < 0.1$ dwarf galaxies (COS-Dwarfs)[23] or the profile seen around the more massive Andromeda Galaxy, M31 (ref. [24]), although large uncertainties in virial radius estimates make such a comparison difficult (see Extended Data Fig. 2b).

Although some sightlines in our sample pass closer to the SMC than the LMC, the common history of the two galaxies should result in a single enveloping Magellanic Corona dominated by the LMC, which is about ten times more massive[8]. Although in isolation SMC-mass galaxies are not massive enough to host their own warm coronae, they

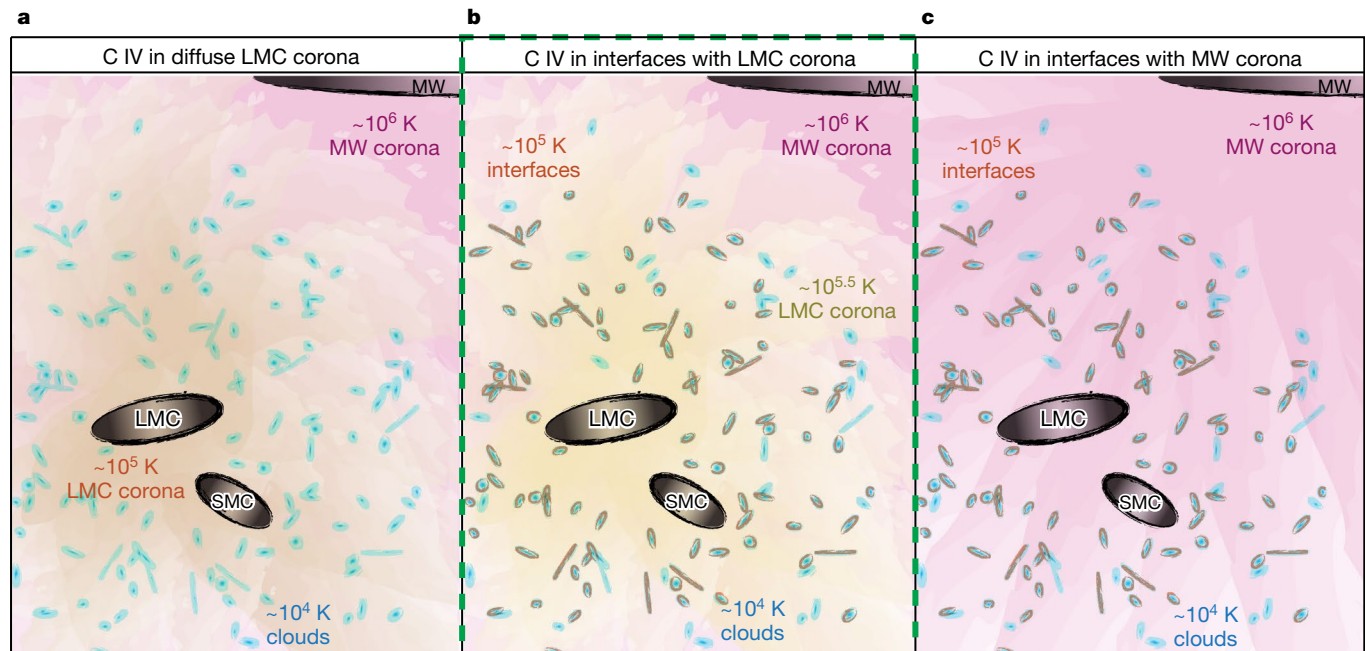

**Fig. 4 | Summary of three possible C IV CGM models.** Our three possible models for C IV in a LMC corona (**a**), interfaces with a LMC corona (**b**), interfaces with a Milky Way (MW) corona (**c**). Cartoon renditions show approximately

$10^4$ K, approximately $10^5$ K, approximately $10^{5.5}$ K and approximately $10^6$ K gas in blue, orange, yellow and pink hues, respectively. The dashed green line outlines model b, which best explains our observations.

can host cool gas in their halos[23]. However, the strong interactions with the LMC and the Milky Way would have strongly disrupted such a cool halo during the infall of the SMC, and so a single Magellanic Corona dominated by the LMC prevails.

The Magellanic Corona should be detectable through its dispersion measure induced in radio observations of extragalactic fast radio bursts[25], as it contains a high column density of free electrons. The presence of such a pervasive corona around the LMC supports the picture of a hierarchical evolution for the Local Group, in which the LMC and SMC accreted onto the Milky Way as part of a larger system of dwarf galaxies, a Magellanic group[26–28], not in isolation. Earlier work has detected the (ultra-faint) galaxies associated with the Magellanic group[29,30]; our evidence for the Magellanic CGM and Corona suggests that we have now detected its gas, an important part of its baryon budget. This provides a more complete understanding of the overlapping and co-evolving ecosystems within the Local Group.

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

## Methods

This work uses archival HST/COS and FUSE spectra to show evidence for the Magellanic Corona. Here we describe our data reduction, Voigt profile fitting and ionization modelling methods. Throughout this work, all reported values and uncertainties are medians and 68% confidence intervals, unless otherwise specified.

### Impact parameters and projection effects

Unlike CGM studies of extragalactic systems, this work focuses on the CGM surrounding the LMC at a distance of only $D = 50$ kpc (ref. [31]). This proximity means that background quasars at large angular separations from the LMC, $\theta$, correspond to relatively small physical separations. The impact parameter, $\rho_{LMC}$, is found using $\rho_{LMC} = D\sin(\theta)$. At $\theta > 45°$, this assumption no longer results in realistic impact parameter estimates and we would require a true 3D model of the location of gas absorbers to calculate a physical separation between gas absorbers and the LMC. Furthermore, for sightlines at large $\theta$, it is harder to kinematically distinguish absorption lines from the LMC and the Milky Way. A larger-scale understanding of the Magellanic Corona and multiphase CGM would only be possible with more reliance on models and simulations to identify the 3D locations of gas absorbers. To keep this work more focused on the observationally derived results, our analysis is thus strictly limited to sightlines within 45° of the LMC, corresponding to 35 kpc in impact parameter.

### HST/COS observations

We design our sample to consist of HST/COS far UV observations of background quasars using both the G130M and G160M gratings, covering the wavelength ranges of about 1,150–1,450 Å and about 1,405–1,775 Å, respectively. The combination of these gratings enable us to examine the following absorption lines: O I $\lambda$1302, N I $\lambda\lambda$1199, 1200, 1200.7, C II $\lambda$1334, Al II $\lambda$1670, Si II $\lambda\lambda$1260, 1193, 1190, 1526, 1304, Si II $\lambda\lambda$ 1250, 1253, 1259, Fe II $\lambda\lambda$ 1608, 1144, Si III $\lambda$1206, C IV $\lambda\lambda$ 1548, 1550 and Si IV $\lambda\lambda$1393, 1402. The COS spectra are processed following previously developed custom reduction and wavelength-calibration methods[10,32] based on the raw products from the calcos[33] data-reduction pipeline. To remove geocoronal airglow contamination in O I $\lambda$1302 and Si II $\lambda$1304, we use a second calcos reduction of the data, using only observations taken during orbital night-time.

The COS far UV observations have a native pixel size of 2.5 km s$^{-1}$ and a spectral resolution (full width at half maximum) of about 20 km s$^{-1}$ and about 15 km s$^{-1}$ for G130M and G160M spectra, respectively. We bin all spectra such that the resulting spectra are Nyquist sampled with two pixels per resolution element.

### FUSE observations

For 15 sightlines in our sample, archival FUSE spectra are also available and analysed to search for O VI $\lambda$1031, 1037 absorption. However, only six sightlines had high enough S/N ratio to make a measurement. These wavelengths fall on the FUSE LIF1 channel with a spectral resolution of about 20 km s$^{-1}$ and native pixel size of 2 km s$^{-1}$, which we bin to Nyquist sample with two pixels per resolution element. These FUSE data are reduced and aligned following the customized methods similar to those used for the HST/COS spectra[34,35]. The O VI $\lambda$1031 may have contamination from molecular $H_2$ absorption at $\lambda$1032.356, which corresponds to roughly 130 km s$^{-1}$ in the O VI frame. However, the expected contribution from this contamination is very small because of the high Galactic latitude of the sightlines and is, in most cases, negligible.

### Absorption-line measurements

We use the open-source Python software, VoigtFit[36], to perform Voigt profile fitting of the absorption in several ions observed with HST/COS with the G130M and G160M gratings. This process uses a least-squares optimizer[37] with recent atomic data[38–40] and convolves the Voigt profile with an approximate instrumental profile of a Gaussian with full width at half maximum corresponding to the observed grating resolution. Although this Gaussian approximation of the instrumental profile is not an exact representation, it has been shown to have a nearly negligible effect on fit results for weak high-velocity components[21]. For all ion fits, we normalize the spectra using a third-order polynomial fit to the continuum surrounding absorption lines of interest. Regions of the absorption spectra that are contaminated by high-redshift absorption components are then flagged to avoid fitting.

We first fit the absorption in all low and intermediate ions (O I, N I, C II, C II*, Si II, Si III, Al II, Fe II) simultaneously, allowing component-line centres to be tied across ions when they show general agreement. The C II* line always contaminates the measurement of absorption in C II at +250 km s$^{-1}$. When there are blended C II components at this velocity, we fix the C II* column density to a constant value of $10^{13.8}$ cm$^{-2}$, based on average measurements from previous work[41], but—in these cases—the measured C II columns near +250 km s$^{-1}$ are not used in our analysis. The Si III $\lambda$1206 transition is frequently saturated, requiring the linewidths to be tied to match the fit Si II linewidths. A minimum allowed linewidth of 9 km s$^{-1}$ is applied on the basis of the instrumental resolution and maximum linewidths are only added as a constraint for highly blended components if they are needed to converge to a best fit.

C IV and Si IV are then fit simultaneously following the same procedure, but independent of the low-ion results to avoid biasing our analysis, because the high-ion component structure may be different. O VI absorption from FUSE is also fit independently when data are available and a reasonable continuum can be determined. If the component structure of the low and high ions match, they are flagged after the fitting process so that their column densities, linewidths and line centres can be compared in the subsequent steps. Furthermore, we calculate upper limits of any transitions in which absorption is not seen on the basis of the S/N ratio of the observed spectra[42,43]. Last, fit components attributed to the Milky Way or known intermediate-velocity or high-velocity clouds are flagged to avoid contaminating our analysis. We note that some contamination from fixed pattern noise persists in our reduced spectra, which may affect our measured column densities and is not accounted for in our estimated errors.

In total, across 28 sightlines, we initially identify 112 unique velocity components that may be attributed to the Magellanic system. We then impose a velocity threshold and only consider absorbers at $v_{LSR} > 150$ km s$^{-1}$ to avoid contamination from absorbers associated with the Milky Way[44]. The velocity threshold of 150 km s$^{-1}$ was determined using a combination of the observed component velocities and simulations of the Magellanic system[7]; it represents the value that best separates the Galactic and Magellanic components and is consistent with previous kinematic studies of Magellanic absorption[12,14]. Furthermore, this velocity threshold is supported by dynamical arguments: given the LMC mass, the virial theorem predicts that Magellanic gas has a velocity dispersion of 50 km s$^{-1}$ centred on the LMC velocity of 280 km s$^{-1}$, implying that 95% of Magellanic gas should be within 180 km s$^{-1}$ and 380 km s$^{-1}$. As a result, our final sample has 52 unique Magellanic velocity components that are further analysed on the basis of their kinematics and photoionization modelling. The Voigt profile model parameters for these 52 C IV and Si IV components are given in Extended Data Table 1 and the ten unique Magellanic O VI absorption components are shown in Extended Data Table 2. Extended Data Fig. 1 shows our measured C IV $\lambda$1548 and O VI $\lambda$1031 absorption-line spectra for our sample. Extended Data Fig. 2a shows the total measured HST/COS column densities in several low and high ions from the Magellanic absorbers at $v_{LSR} > 150$ km s$^{-1}$ as a function of the LMC impact parameter. All low ions show a declining radial profile, similar to the relation shown in the high ions (Fig. 2).

A comparison of our observed radial profile with that seen in the COS-Dwarfs survey[23] and M31 (ref. [24]) is shown in Extended Data Fig. 2b. We normalize impact parameter measurements across these surveys

based on the radius enclosing a mean overdensity of 200 times the critical density, $R_{200}$, which is often used as a measure of the virial radius in CGM studies. In the radial region of overlap between these surveys and our work, the declining profile of the LMC is more concentrated, with a possibly truncated profile. Because the LMC halo is already within the virial radius of the Milky Way, it is expected to be tidally truncated, hence such a truncated profile is expected. However, the uncertainties in estimates of $R_{200}$ are estimated to be 50% in the COS-Dwarfs and M31 surveys, with the LMC value we use at $R_{200} = 115 \pm 15$ kpc.

Our spectra can be accessed publicly on the Barbara A. Mikulski Archive for Space Telescopes (MAST). A full table of our fit parameters, along with summary plots of our best fits, can be accessed on GitHub at https://github.com/Deech08/HST_MagellanicCorona.

## Ionization models

We use 1D Cloudy[45] radiative transfer models to simulate the physical conditions of the absorbing gas. Our Cloudy models require four key inputs to run: (1) an external radiation field, (2) the observed column density measurements, (3) a specified stopping condition to reach for convergence and (4) a gas-phase metallicity. All models assume a plane-parallel geometry and constant gas density.

Incident radiation fields in Cloudy require a shape and intensity. We adopt the Milky Way escaping radiation field model to set the shape of the radiation field, assuming that the radiation fields from the LMC and the SMC have the same spectral shape[10,46,47]. The intensity of the radiation field towards each sightline is set by a hydrogen-ionizing photon flux $\Phi_H$ determined from published ionization models, which includes contributions from the LMC, the SMC and the Milky Way[20]. We reconstruct this model in 3D space to interpolate an initial value for $\Phi_H$ for any specified location. In our model, we allow $\Phi_H$ to be a free parameter, because a precise distance to the absorbing material is not known. We also include a constant contribution from an extragalactic UV background[48] and cosmic ray background[49].

We use Cloudy's built-in 'optimize' command to vary our free parameters and find optimal parameters to explain our observed column densities and upper limits[45,50]. The optimize models use up to three possible free parameters: (1) the hydrogen-ionizing photon flux, $\Phi_H$, described above, (2) the total hydrogen number density, $n_H$, which is the sum of the ionic, atomic and molecular hydrogen densities of the plasma that is to be modelled, and (3) the neutral hydrogen column density ($N_{HI}$) stopping condition. For sightlines with an H I or O I detection, the observed H I or O I column-density measurement serves as the stopping condition and the model only uses the first two free parameters ($\Phi_H$ and $n_H$). For sightlines without an H I or O I, we use all three free parameters ($\Phi_H$, $n_H$ and $N_{HI}$). Once Cloudy's optimize method has found a possible solution of parameters, we run one final Cloudy model at the specified optimal parameters to produce predictions of ion column densities and gas temperatures, including predictions for high-ion (Si IV, C IV, O VI) column densities. To ensure Cloudy does not settle at local minima in the optimization process, we use a broad range of initial densities from $\log 10(n_H/\mathrm{cm}^{-3}) = -3$ to 1 and ionizing fluxes $\Phi_H$ in a range of 3 dex around the model prediction at $D = 50$ kpc, but still find a resulting narrow range of total hydrogen densities ($n_H$), ionized gas temperatures ($T_e$), neutral atomic hydrogen columns ($N_{HI}$) and ionized-to-neutral atomic hydrogen ratios $N(\mathrm{H\,II})/N(\mathrm{H\,I})$ across all sightlines and velocity components. Furthermore, we have also run a coarse grid at a larger range of free parameters to help confirm that our solutions are indeed optimal and not local minima.

Although interstellar medium gas-phase metallicities have been measured in the LMC, SMC, Magellanic Bridge[51] and Magellanic Stream[52–55], the metallicity of the Magellanic CGM is highly uncertain. To estimate the gas metallicity, we use a sightline in our sample towards HE 0226−4110 that overlaps with recently published analysis of FUSE spectra to measure neutral hydrogen column densities[56]. Two absorption components towards this sightline may belong to the Magellanic

Corona at $v_{LSR} = +174$ km s$^{-1}$ and $+202$ km s$^{-1}$, providing a measured neutral hydrogen column density to set as a stopping condition in Cloudy. Unfortunately, there is no detected O I absorption in either the COS or the FUSE data, so a metallicity is calculated using a Cloudy optimize model (described above), allowing the total hydrogen density, hydrogen-ionizing photon flux and metallicity to vary. The Cloudy models are optimized on the basis of the measured COS column densities across all available metal ions and any upper limits when absorption is not detected. The results for these two components are $\log_{10}(\Phi_H/\mathrm{photons\ s}^{-1}) = 5.06$, $\log_{10}(n_H/\mathrm{cm}^{-3}) = -1.58$ and [Z/H] = $-0.72$ for the $v_{LSR} = +174$ km s$^{-1}$ component and $\log_{10}(\Phi_H/\mathrm{photons\ s}^{-1}) = 4.95$, $\log_{10}(n_H/\mathrm{cm}^{-3}) = -1.91$ and [Z/H] = $-0.62$ for the $v_{LSR} = +202$ km s$^{-1}$ component. On the basis of these results, we adopt the average [Z/H] = $-0.67$ as the gas-phase metallicity for photoionized gas. For hotter gas in interfaces and the corona, we assume a gas-phase metallicity of [Z/H] = $-1$, because we expect this more primordial gas to be at lower metallicity.

Our optimal set of Cloudy models provides predictions for the expected column densities of the high ions Si IV, C IV and O VI for a single-phase photoionized gas. However, the observed high-ion columns are much greater (by orders of magnitude) than the photoionization predictions. Across all sightlines and absorption components that may be associated with the Magellanic system, we find that 72% of Si IV and 84% of C IV absorption components are less than 10% photoionized. We use this 10% (1-dex) threshold to define our sample of Magellanic absorbers that are not photoionized (see shaded components in Fig. 2). These C IV and Si IV absorbers probably arise in interfaces in the range $T = 10^{4.3–4.9}$ K.

The observed triply ionized Magellanic absorption is well described using either equilibrium or time-dependent non-equilibrium collisional ionization models[16]. In both cases, we can infer an electron temperature based on the ratio of C IV and Si IV column densities, because the close similarity of the C IV and Si IV line profiles indicates that the two ions are co-spatial. The modelled relation of this column-density ratio with temperature for the equilibrium model and for isobaric and isochoric time-dependent models is shown in Extended Data Fig. 3 for a range of metallicities. The inferred temperature is then used to determine a C IV ionization fraction, from which the total ionized hydrogen (H II) column density can be calculated, resulting in the measurements shown in Fig. 3b. In total, the temperature distributions of the photoionized and collisionally ionized gas are shown in Extended Data Fig. 4b. In the sightlines in which we have measured O VI absorption, we find that the O VI absorbing gas requires a higher temperature than the C IV and Si IV absorbing gas, indicating that the O VI arises in a separate, hotter phase. Whereas at high metallicity lower-temperature solutions for our observed column-density ratios are possible, this is not the case at the lower metallicities (below 0.1 solar) expected for Magellanic coronal gas.

We also consider more recent collisional ionization models that include photoionization from an extragalactic background[57]. However, these models do not include the non-isotropic radiation fields necessary for modelling clouds near the Milky Way and the LMC, and only offer approximate predictions using a general background radiation field. Instead, we only consider the two cases of entirely photoionized or entirely collisionally ionized in this work, but note that a full picture will require considering collisional ionization and photoionization from the Milky Way and the Magellanic Clouds together.

## Statistical significance of results

Here we describe the statistical tests we used to support our claims of significance. Throughout this work, we adopt a significance threshold $P$-value of $P = 0.05$.

**Velocity structure.** In our Voigt profile fitting process, individual components are initially paired across low ions and high ions based on their approximate centroid velocities. This pairing process is inherently

biased, as it assumes that components across ions are physically tied and results in the lowest possible differences in velocity centroids for our analysis. However, for the low and high ions, the velocity structure was qualitatively well matched to one another, with absorption components at similar velocities for both cases. This correspondence is less clear for the O VI absorption line centroids, so matching O VI components in the same manner is much more uncertain. Combined with the relatively low S/N ratio (≈10) and moderate velocity resolution (20 km s$^{-1}$) of our spectra, we are unable to fully resolve all absorption components. We therefore find that comparisons of the kinematic properties of low and high ions are generally inconclusive. However, the kinematics are still consistent with our primary conclusion that C IV and Si IV arise in the interfaces between cool clouds and a Magellanic Corona, because in an interface model the velocity structure of the low ions and the high ions should be linked. When considering O VI, we calculate the velocity offset from the closest absorption component in other ions (Si III or C IV) and find that the widths of the velocity-offset distributions have standard deviations of $\sigma_{\text{OVI–SiIII}} = 22^{+7}_{-4}$ km s$^{-1}$ and $\sigma_{\text{OVI–CIV}} = 22^{+10}_{-6}$ km s$^{-1}$, respectively. This is $7^{+12}_{-8}$ km s$^{-1}$ greater in width of the distribution of velocity differences between the low ions and C IV matched in the same manner, supporting the result that O VI exists in a different phase.

**Linewidths.** We show the paired (matched on the basis of their velocities during the Voigt profile fitting process) differences of component linewidths in Extended Data Fig. 5a. Differences in paired linewidths do not show statistical significance. However, when considering our populations of linewidth measurements, we do find a statistically significant difference between the linewidth distributions of singly ionized C and Si in comparison with triply ionized C and Si (see their distributions in Extended Data Fig. 4c,d). The Anderson–Darling statistical test of the null hypothesis that the singly and triply ionized linewidths are drawn from the same underlying population can generally be rejected at the $P$-value threshold of 0.05 for both C and Si. We perform the test on 10,000 bootstrap samples to account for measurement errors of linewidths. The C IV and C II linewidths return a $P$-value (with 68% confidence intervals) of $P_\text{C} = 0.008^{+0.08}_{-0.007}$, with 78% of bootstrap samples below our $P$-value threshold of 0.05. Similarly, the Si IV and Si II linewidths return $P$-values of $P_\text{Si} = 0.001^{+0.01}_{-0.0}$, with 93% of bootstrap samples below our significance threshold.

**Declining radial profile.** We test the statistical significance of the anti-correlation between the C IV and Si IV with LMC impact parameter using Kendall's $\tau$ rank correlation coefficient with censoring, which provides a robust measure of the monotonic relationship between two variables[58,59]. The Magellanic Corona shows a distribution of coefficients that are negative for both C IV and Si IV, with mean values of $\tau = -0.4 \pm 0.1$ and $\tau = -0.3 \pm 0.1$, respectively, as shown in Extended Data Fig. 6. The $P$-values for C IV allow the null hypothesis of no correlation to be rejected at the 0.05 level for 97% of bootstrap samples, whereas the $P$-values for Si IV can only be rejected for 73% when considering all of our data. When only considering the absorbers at $\rho_\text{LMC} > 7$ kpc, the significance of the Si IV anti-correlation becomes stronger, with $P < 0.05$ for 89% of 10,000 bootstrap samples and a mean value of $\tau = -0.4 \pm 0.1$, but the change for C IV is negligible. The best-fit lines for the anti-correlation are found using a Markov chain Monte Carlo analysis with censoring to account for upper limits and measurement errors[60]. For the O VI measurements, Kendall's $\tau$ rank correlation coefficient is less reliable, as we only have six data points, and is not conclusive.

**Magellanic Corona versus tidally stripped stream with interfaces**
Previous simulations have been able to explain much of the ionized gas associated with the Magellanic Stream by tidal stripping, without the presence of a corona[61]. If this were the case, and the Stream were the

dominant source of ionized gas, we would expect to see a stronger correlation of C IV column density as a function of distance from the Magellanic Stream (absolute Magellanic Stream latitude) than as a function of the LMC impact parameter. We use the partial Spearman rank-order correlation test to assess the strength of the correlation between our measured ion column densities and either the LMC impact parameter or the absolute Magellanic Stream latitude, while removing the effects of the other. We note that, for this test, we are only considering the collisionally ionized C IV and Si IV columns, but considering all the observed columns for low ions. The correlation coefficients and $P$-values of the test with a null hypothesis of no correlation are given in Extended Data Table 3. For most ions, the correlation is substantially stronger with the LMC impact parameter, after removing the effects of the absolute Magellanic Stream latitude. However, the partial correlation test for Fe II is inconclusive and the test for O I suggests a stronger correlation with the absolute Magellanic Stream latitude. These tests are consistent with a Magellanic Corona and CGM origin to the gas absorbers we have measured, with the exception of O I, which may be more biased towards tracing cooler, tidally stripped gas in the Magellanic Stream.

In Extended Data Fig. 7, we show our measurements of collisionally ionized C IV columns on a map of the Magellanic system in Magellanic coordinates, alongside measurements of all C IV absorption from a previous survey of the Magellanic Stream[10]. When considering all C IV, the surface density profile is much more extended along the direction of the Magellanic Stream, but with our adopted velocity threshold and removal of photoionized gas, the radial profile centred on the LMC is apparent, especially when considering sightlines that overlap on our sample and the previous sample.

In this previous work, much of the observed C IV absorption was interpreted to arise from interfaces around the tidally stripped, cooler gas from the LMC with a hot, approximately $10^6$ K Milky Way corona. The basic premise of this conclusion is still valid in our sample, but the strong radial profile centred on the LMC suggests that the hotter gas interacting to form the interfaces should also be centred on the LMC, not the Milky Way. Therefore, a Magellanic Corona at roughly $10^{5.5}$ K can explain our observed radial profile and the observed C IV absorption.

**Mass estimates**
Our estimates of the mass for each phase of the Magellanic CGM are derived from the relation between the ionized hydrogen column density and the LMC impact parameter. For each phase (approximately $10^4$ K, approximately $10^{4.9}$ K and approximately $10^{5.5}$ K), a best-fit linear regression model is fit to the ionized hydrogen column as a function of $\rho_\text{LMC}$. Then the ionized hydrogen mass in each phase is calculated using

$$M_\text{H II} = \int_{0 \text{ kpc}}^{35 \text{ kpc}} N_\text{H II}(\rho_\text{LMC}) m_\text{p} \, 2\pi\rho_\text{LMC} f_\text{cov} \, \text{d}\rho_\text{LMC} \tag{1}$$

in which $m_\text{p}$ is the proton mass and $f_\text{cov}$ is the covering fraction.

For the approximately $10^4$ K gas, the ionized hydrogen column density in each direction is derived directly from the Cloudy models, with a covering fraction $f_\text{cov} = 0.82$, as low ions are detected at Magellanic velocities in 23/28 directions in our sample. However, we note that the covering fraction of low ions tends to decrease as a function of the LMC impact parameter, but use a constant covering fraction as an approximation.

For the $10^{4.9}$ K gas, the total ionized hydrogen column density in each sightline is derived on the basis of the C IV column density and best-fit temperature from the collisional ionization models[16] using

$$N_\text{H II} = \frac{N_\text{C IV}}{f_\text{C IV} \, [\text{Z/H}]}, \tag{2}$$

in which $f_\text{C IV} \equiv \text{C}^{3+}/\text{C}$ is the fraction of triply ionized carbon at the best-fit temperature and $[\text{Z/H}] = 0.21[\text{Z/H}]_\odot$ is the metallicity. For C IV, the

covering fraction is set to $f_{cov} = 0.78$ for $\rho_{LMC} < 30$ kpc and $f_{cov} = 0.3$ for $\rho_{LMC} \geq 30$ kpc based on the observed detection rate of C IV absorption in our sample. The relation between the derived column densities and the LMC impact parameter allow for our mass calculations for the approximately $10^4$ K and approximately $10^{4.9}$ K gas to converge, changing by at most 0.1 dex if instead we integrate out to 500 kpc.

For the approximately $10^{5.5}$ K gas, the mass is again found on the basis of the O VI absorption columns in the collisional models, using

$$N_{HII} = \frac{N_{OVI}}{f_{OVI}\,[Z/H]}, \tag{3}$$

and using the same covering fraction correction used for the approximately $10^{4.9}$ K gas. Here we use the maximal $f_{OVI}$ value for each of the collisional models, which peak near $10^{5.5}$ K at $f_{OVI} \approx 0.2$. The best-fit line for this phase does not converge, so integrating the radial profile depends highly on the radial range considered. Instead, we only integrate between the bounds of our observations (6.7 kpc < $\rho_{LMC}$ < 32.5 kpc) and present an approximate corona mass for this region only.

## Data availability

HST/COS spectra used in this work are publicly available on the MAST at https://archive.stsci.edu/. These archival observations can be found under the following HST program IDs: 11692, 15163, 12263, 11686, 11520, 12604, 12936, 11541 and 14655.

## Code availability

Voigt profile fit results, summary fit spectra and custom code used can be found in our GitHub repository at https://github.com/Deech08/HST_MagellanicCorona. Furthermore, the following software was used in this work: Astropy[62,63], calcos[33], cartopy[64], lmfit[37], SciPy[65], VoigtFit[36], Cloudy[45] and Pingouin[66].

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

**Acknowledgements** This work was funded through HST Archival Program 16363, provided by NASA through a grant from the Space Telescope Science Institute, which is operated by the Association of Universities for Research in Astronomy, Inc., under NASA contract NAS5-26555. This work uses observations made with the NASA/ESA Hubble Space Telescope, obtained from the Data Archive at the Space Telescope Science Institute, which is operated by the Association of Universities for Research in Astronomy, Inc., under NASA contract NAS5-26555. D.K. is supported by an NSF Astronomy and Astrophysics Postdoctoral Fellowship under award AST-2102490.

**Author contributions** D.K. led the investigation, formal analysis, methodology, visualization and writing. A.J.F. led the project development and management and contributed heavily to the project inception, funding and writing. E.D.O. led the conceptualization and is Principal Investigator of the Hubble Space Telescope grant that funded the research. B.P.W. led the data curation and contributed heavily to the proposal writing. A.J.F., E.D.O. and B.P.W. contributed to validation, methodology and reviewing and editing. D.M.F. contributed to data curation. F.H.C. contributed to methodology and visualization. S.L., F.H.C., D.M.F., J.C.H. and N.L. contributed to validation and reviewing and editing.

**Competing interests** The authors declare no competing interests.

**Additional information**
**Correspondence and requests for materials** should be addressed to Dhanesh Krishnarao.

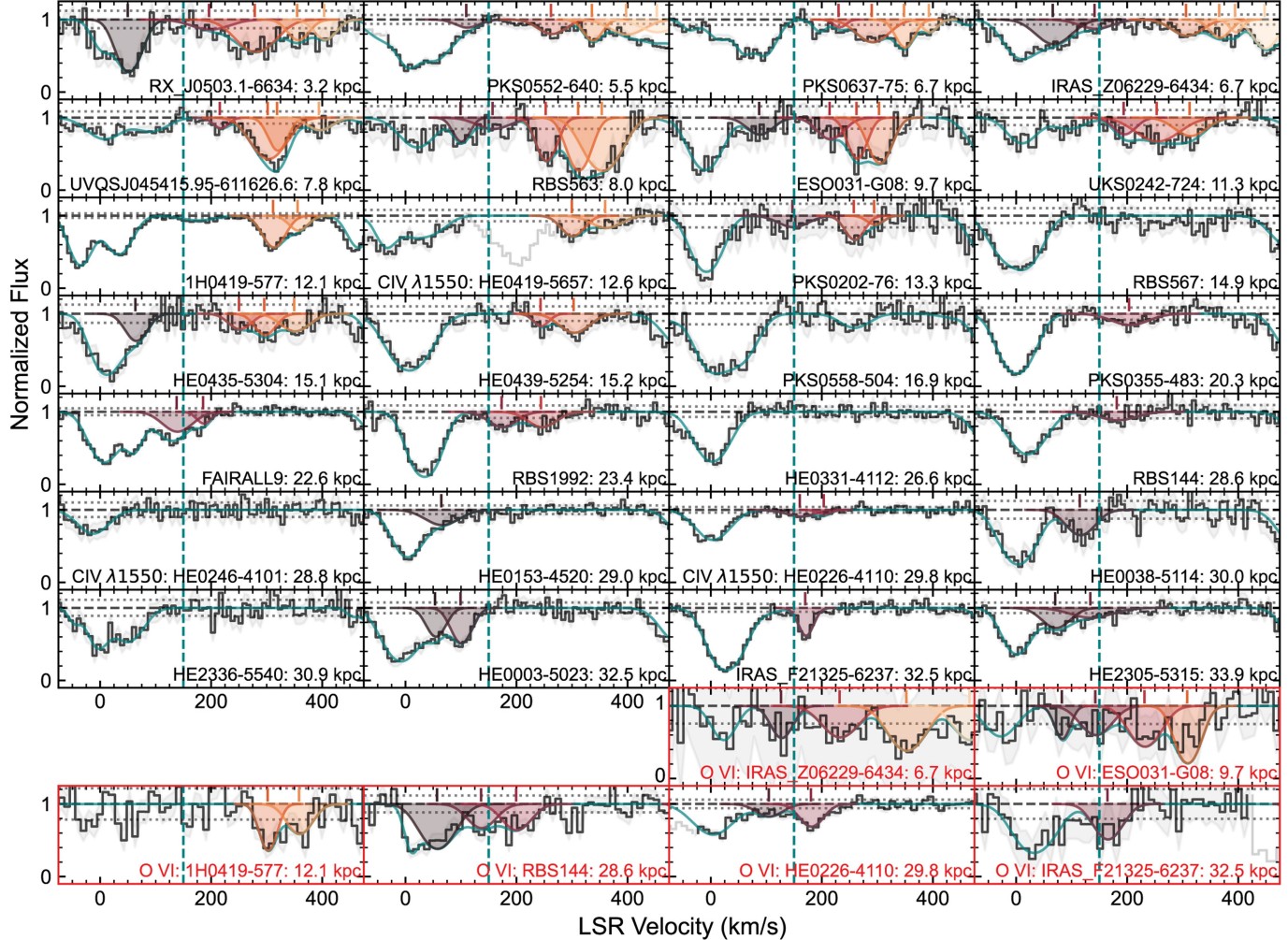

**Extended Data Fig. 1 | Sample absorption-line spectra.** Normalized HST/COS spectra of the C IV λ1548 (upper panels) and O VI λ1031 (lower panels, red outlines) absorption lines in the LSR velocity frame, ordered by their LMC impact parameters (low to high). The normalized flux is shown in black, with 1σ uncertainties shaded in grey around them. The solid teal line shows the full Voigt profile fit to the C IV λ1548, 1550 and O VI λ1031, 1037 doublets, with individual Magellanic components shaded in red-orange hues, corresponding to the same colour scheme used in Fig. 1. Component centres are shown with tick marks. The dashed vertical line marks the 150 km s⁻¹ threshold used in this work. Greyed-out portions of the spectra are contaminated by higher-redshift absorbers and are not considered in our analysis. C IV λ1550 spectra are shown in cases in which the C IV λ1548 is highly contaminated. Further fit results for all ions and sightlines can be viewed at https://github.com/Deech08/HST_MagellanicCorona.

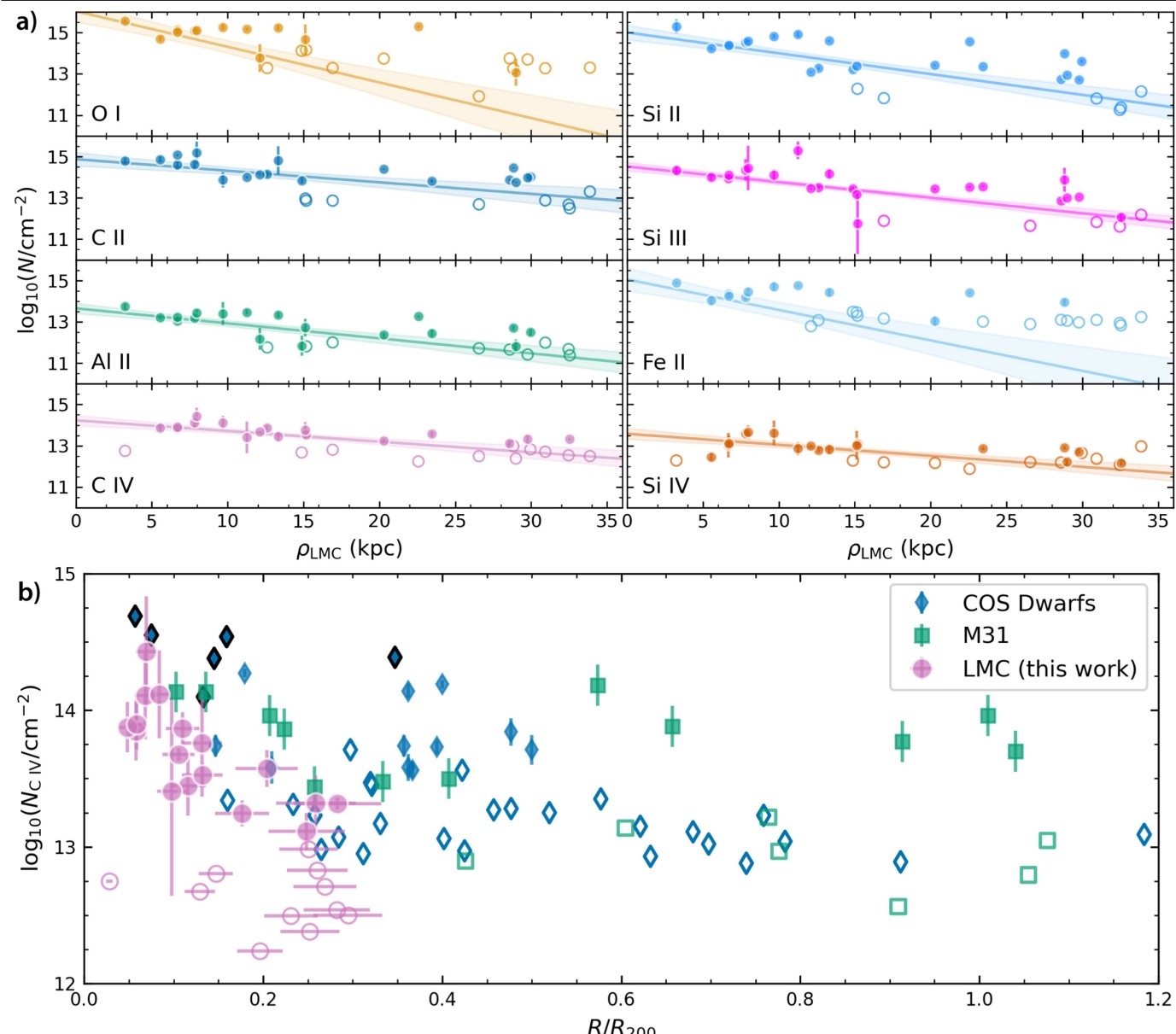

**Extended Data Fig. 2 | Radial profiles of ionic column densities.** Measured column densities of Magellanic components at $v_{LSR} > 150$ km s$^{-1}$ as a function of the LMC impact parameter, $\rho_{LMC}$, with open circles marking upper limits in **a**. For each ion, the best-fit line and uncertainties are found using a Markov chain Monte Carlo analysis with censoring, as in Fig. 2, for all impact parameter for low ions and only $\rho_{LMC} > 7$ kpc for C IV and Si IV. Each panel corresponds to the ion labelled in the lower left. **b**, A comparison of C IV column densities as a function of impact parameter normalized by the radius enclosing a mean overdensity of 200 times the critical density, $R_{200}$, of the host galaxy with the COS-Dwarfs survey[23] and M31 (ref. [24]). The LMC data use $R_{200} = 115 \pm 15$ kpc.

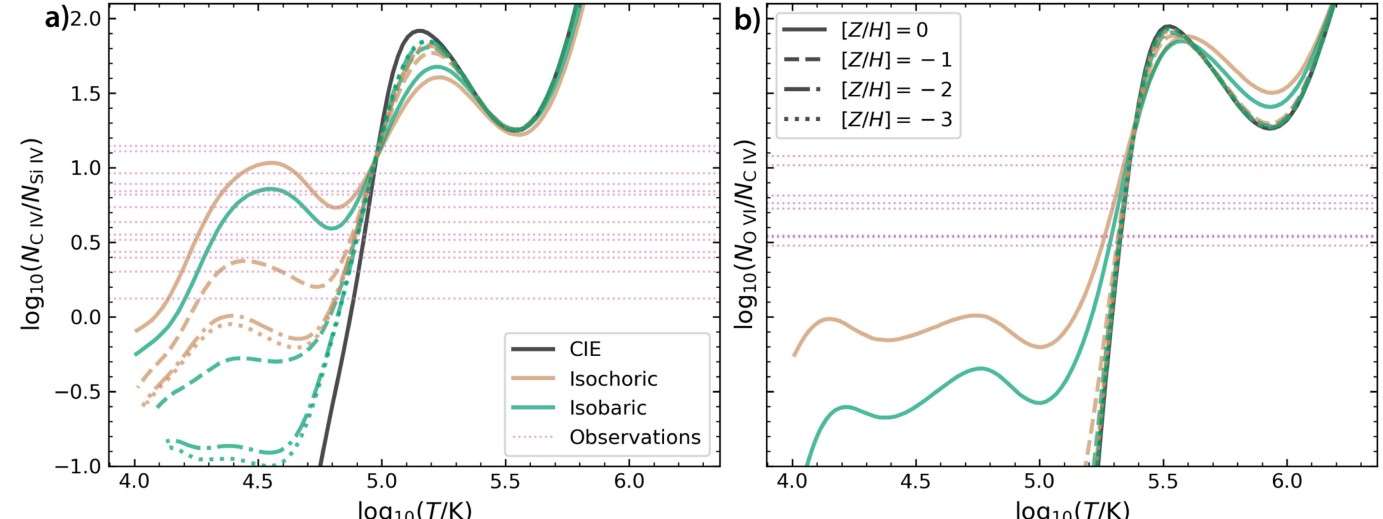

**Extended Data Fig. 3 | Collisional ionization model relations.** Predicted column-density ratios as a function of temperature based on the radiative cooling gas models in collisional ionization equilibrium (CIE; black; **a**), time-dependent isochoric cooling (brown; **b**) or time-dependent isobaric cooling (green; **b**)[16]. Dotted pink lines show observed values of column-density ratios in the Magellanic components. Solid, dashed, dot-dashed and dotted lines correspond to metallicities of [Z/H] = 0, −1, −2 and −3, respectively.

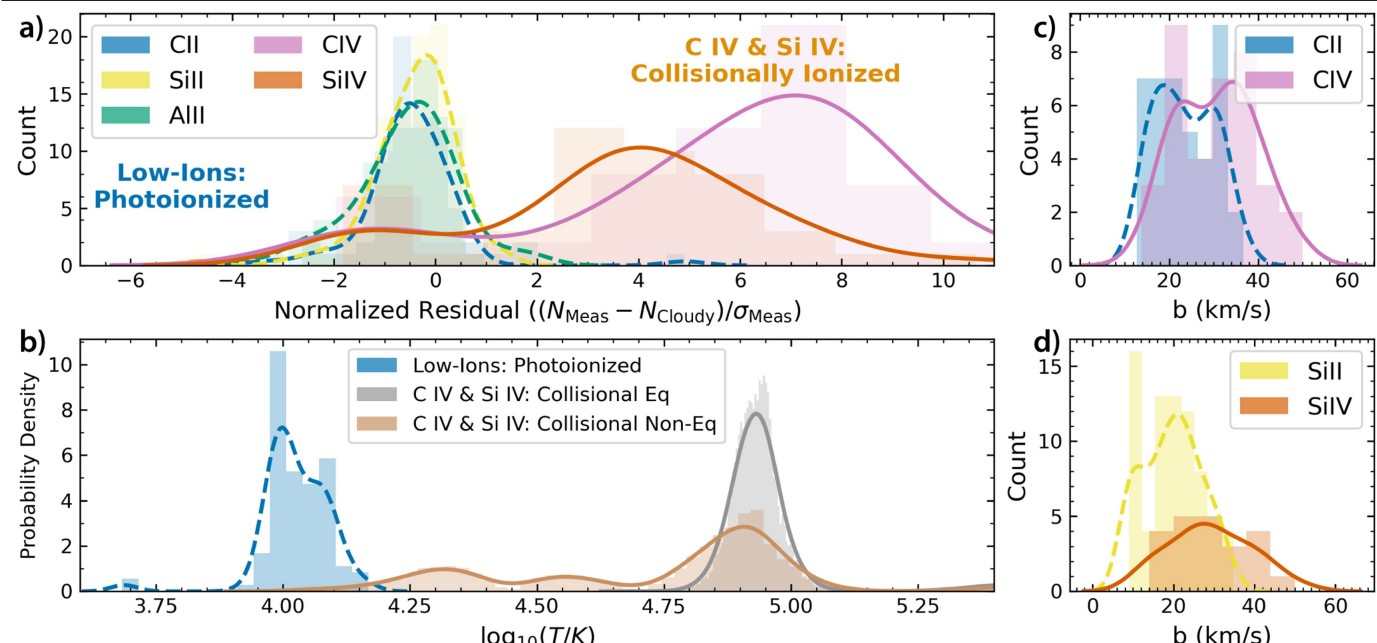

**Extended Data Fig. 4 | Evidence for a multiphase Magellanic CGM.**
**a**, Histograms and Gaussian kernel density estimates (KDEs) of the residuals between predicted ion column densities from Cloudy photoionization models and measured ion column densities, normalized by their standard deviation uncertainties. The histograms are shown for low ions (C II, Si II and Al II; dashed lines in blue, yellow and green, respectively) and high ions (C IV and Si IV; solid lines in pink and orange, respectively). **b**, The inferred gas temperatures for the photoionized gas (blue; $\log_{10}(T_{\mathrm{e}}/\mathrm{K}) = 4.02^{+0.07}_{-0.04}$) and the collisionally ionized gas under equilibrium (grey; $\log_{10}(T_{\mathrm{e}}/\mathrm{K}) = 4.92^{+0.05}_{-0.02}$) and non-equilibrium isochoric (brown; $\log_{10}(T_{\mathrm{e}}/\mathrm{K}) = 4.87^{+0.09}_{-0.06}$; high-temperature solution only) models[16]. **c**,**d**, measured linewidths for C II and C IV absorption (upper; **c**) and Si II and Si IV absorption (lower; **d**).

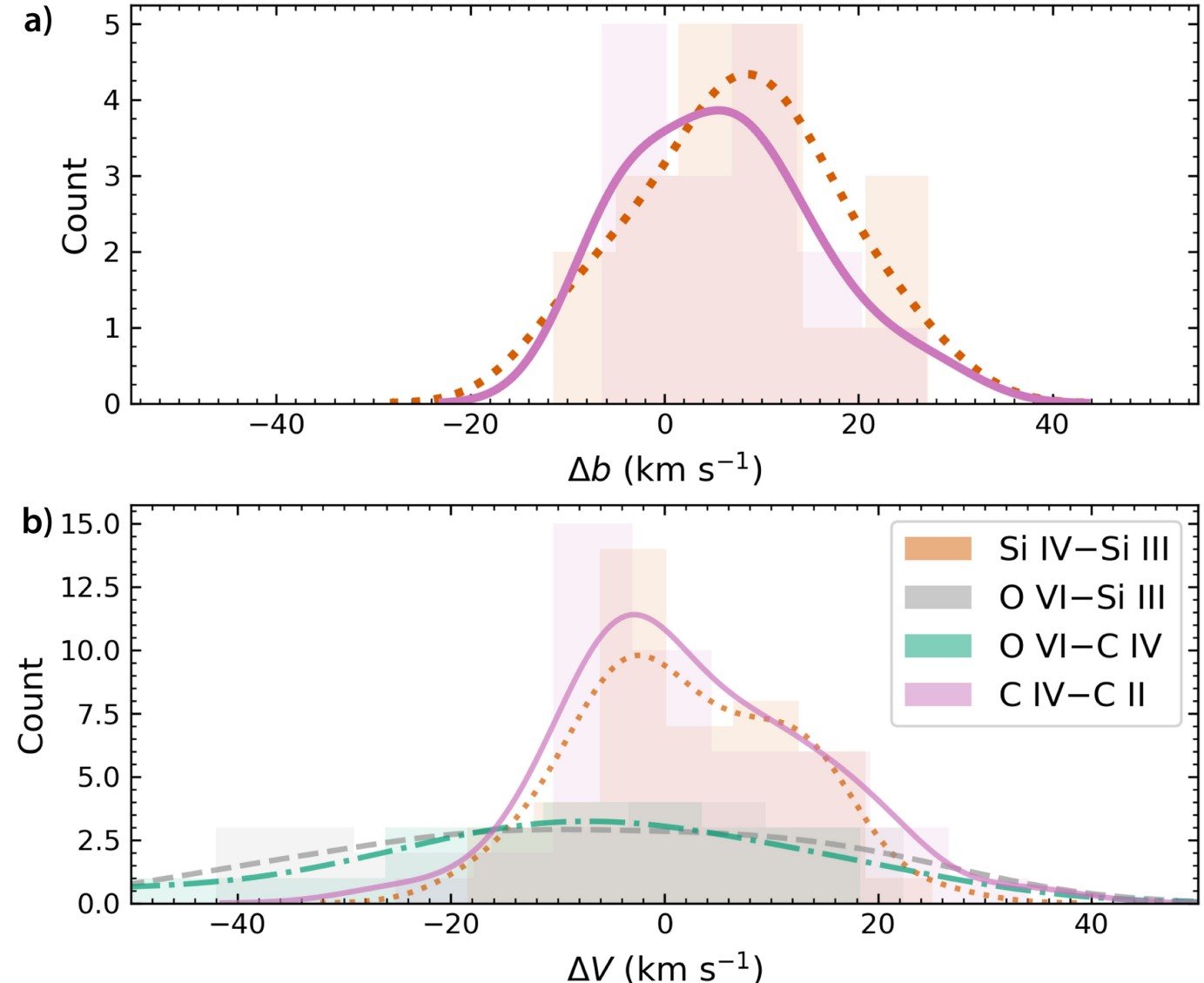

**Extended Data Fig. 5 | Kinematic differences of paired high and low ions.** Histograms and Gaussian KDEs of pairs of high–low-ion linewidths (**a**) and velocity centroids (**b**), separated by C ions (pink; solid line) and Si ions (orange; dotted line). The velocity centroid panel also includes differences between O VI and either Si III (grey; dashed line) or C IV (green; dot-dashed line). The paired linewidths undergo a one-sided Wilcoxon signed-rank test, whereas the velocity centroids undergo a two-sided test.

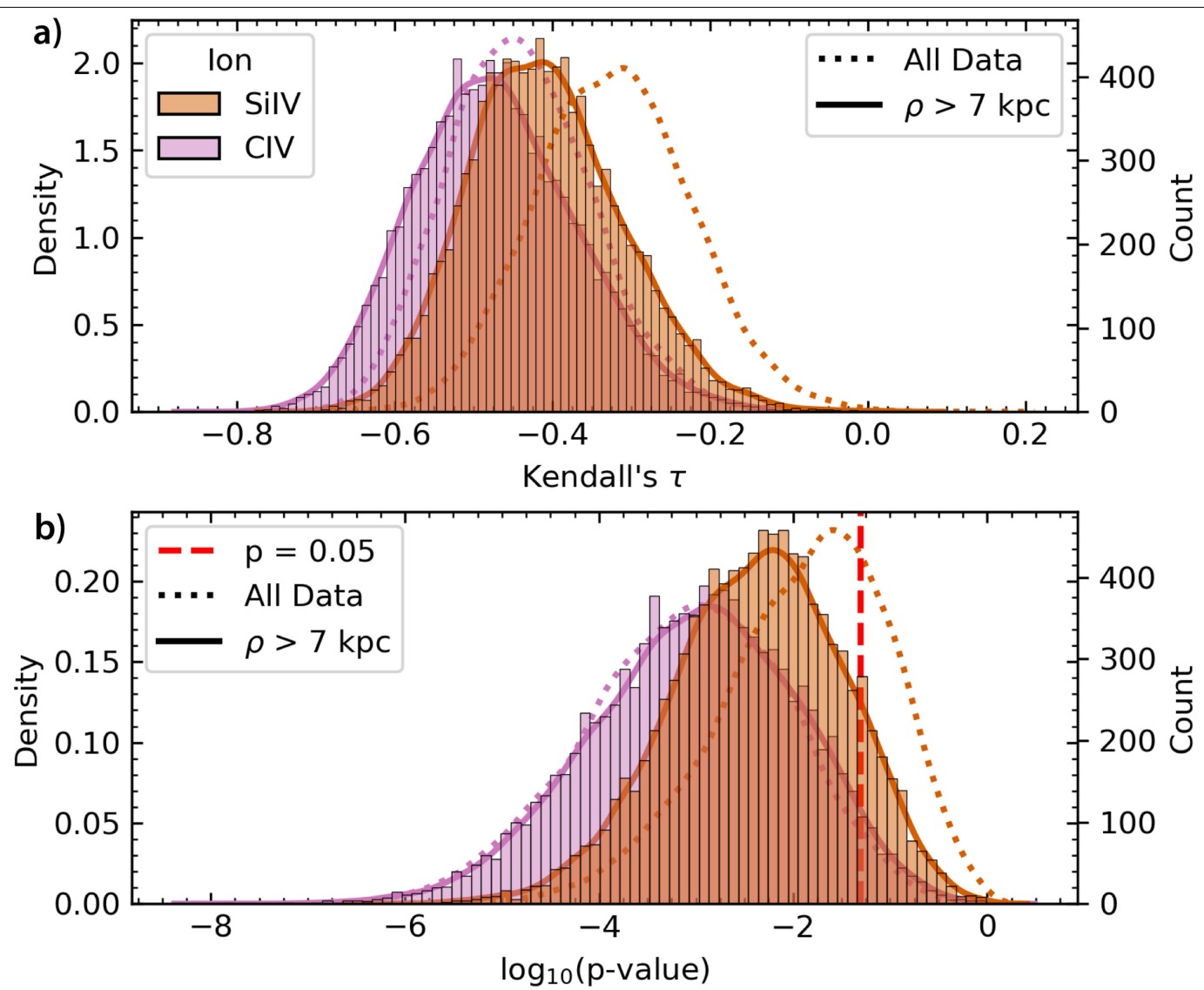

**Extended Data Fig. 6 | Statistical significance of radial profile.** Results of a correlation analysis. Kendall's $\tau$ correlation test with censoring[58] is performed using 10,000 bootstrap samples, with the resulting distributions of test statistics (**a**) and $P$-values (**b**) shown for the null hypothesis of no correlation. The dotted line shows the Gaussian KDE for the distribution using all data points, whereas the solid line and shaded histograms show the result for only data at $\rho_{LMC} > 7$ kpc.

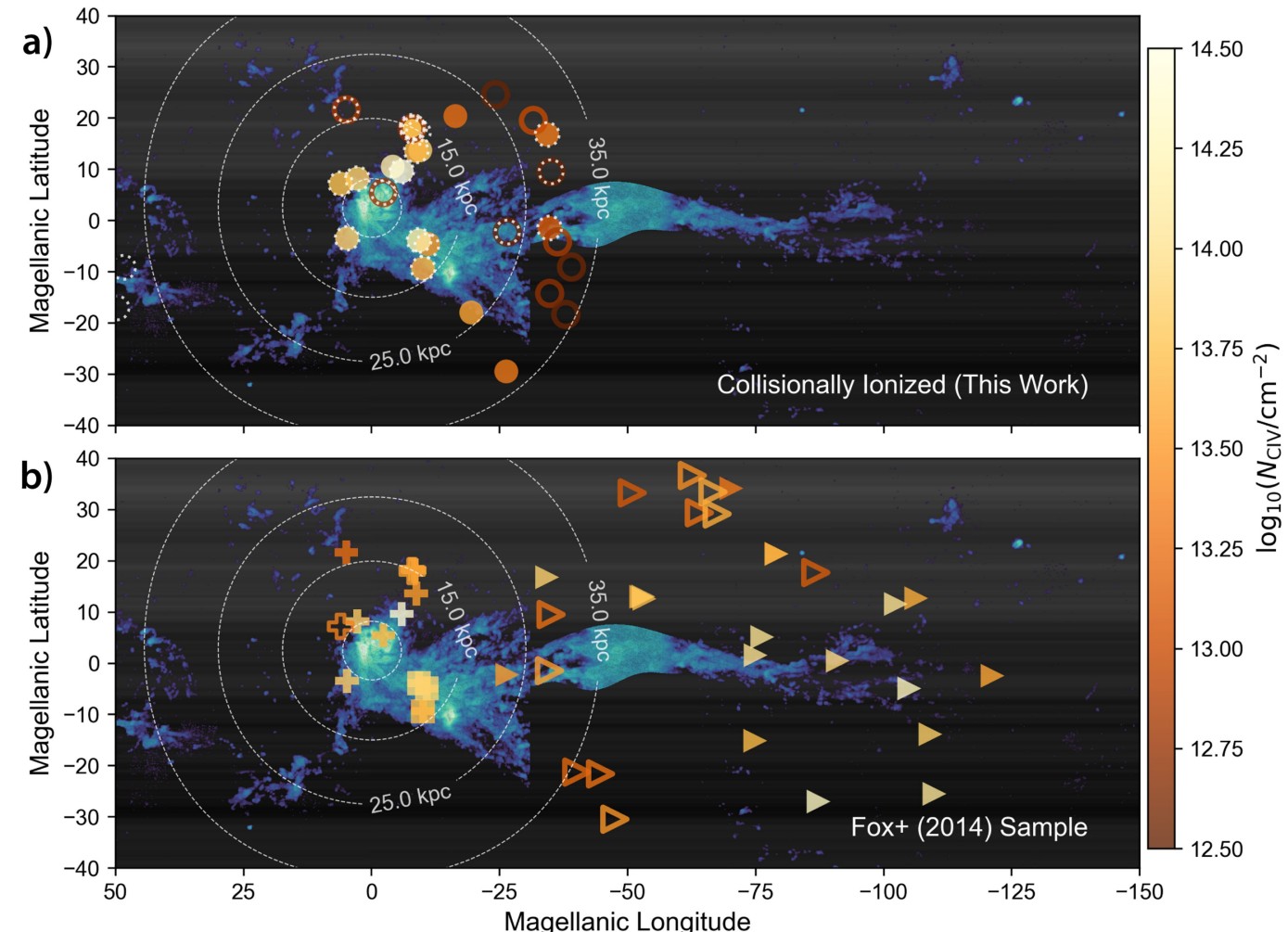

**Extended Data Fig. 7 | Magellanic Corona sample versus Magellanic Stream sample.** Map of measured C IV column density shown in Magellanic coordinates of our collisionally ionized Magellanic sample (**a**; photoionized phase removed) and previous measurements of all C IV absorption in a survey of the Magellanic Stream[10] (**b**). Squares and triangles in the lower panel mark sightlines that were flagged as belonging to the Magellanic Bridge and Stream, respectively, whereas other sightlines are marked with a plus symbol. Open symbols show upper limits. Symbols with a dotted white outline in the upper panel denote sightlines that have overlap with the previous survey[10], whereas those without a white outline are unique to this work.

## Extended Data Table 1 | Voigt profile model parameters for the Magellanic CGM

| Source Name | RA [deg] | Dec [deg] | $\rho_{LMC}$ [kpc] | $v_{C\,IV}$ [km s$^{-1}$] | $b_{C\,IV}$ [km s$^{-1}$] | $\log_{10}\left(\frac{N_{C\,IV}}{cm^2}\right)$ | $f_{PI,C\,IV}$ | $v_{Si\,IV}$ [km s$^{-1}$] | $b_{Si\,IV}$ [km s$^{-1}$] | $\log_{10}\left(\frac{N_{Si\,IV}}{cm^2}\right)$ | $f_{PI,Si\,IV}$ | $\log_{10}\left(\frac{N_{C\,IV}}{N_{Si\,IV}}\right)$ |
|---|---|---|---|---|---|---|---|---|---|---|---|---|
| RX J0503.1-6634 | 75.77 | −66.56 | 3.2 | 196 ± 18 | 45 ± 27 | 13.20 ± 0.31 | 6.05 ± 4.30 | 196 ± 18 | 42 ± 20 | 12.87 ± 0.22 | 5.93 ± 3.05 | 0.33 ± 0.38 |
| RX J0503.1-6634 | 75.77 | −66.56 | 3.2 | 279 ± 6 | 45 ± 12 | 13.80 ± 0.10 | 1.54 ± 0.35 | 279 ± 6 | 39 ± 15 | 13.14 ± 0.17 | 3.21 ± 1.29 | 0.66 ± 0.20 |
| RX J0503.1-6634 | 75.77 | −66.56 | 3.2 | 355 ± 8 | 23 ± 13 | 13.30 ± 0.25 | 4.80 ± 2.75 | 355 ± 8 | 40 ± 11 | 12.97 ± 0.14 | 4.66 ± 1.47 | 0.33 ± 0.28 |
| RX J0503.1-6634 | 75.77 | −66.56 | 3.2 | 404 ± 9 | 24 ± 11 | 13.29 ± 0.19 | 4.08 ± 1.78 | — | — | < 12.29 | — | > 1.01 |
| PKS0552-640 | 88.10 | −64.04 | 5.5 | 397 ± 6 | 40 | 13.34 ± 0.26 | < 0.01 | 397 ± 6 | 9 | 12.45 ± 0.19 | 0.03 ± 0.01 | 0.89 ± 0.32 |
| PKS0552-640 | 88.10 | −64.04 | 5.5 | 336 ± 3 | 19 ± 5 | 13.24 ± 0.15 | < 0.01 | — | — | < 12.29 | — | > 0.96 |
| PKS0552-640 | 88.10 | −64.04 | 5.5 | 262 ± 4 | 32 ± 7 | 13.24 ± 0.07 | < 0.01 | — | — | < 11.99 | — | > 1.25 |
| PKS0552-640 | 88.10 | −64.04 | 5.5 | 453 ± 12 | 40 ± 14 | 13.26 ± 0.24 | < 0.01 | — | — | < 12.28 | — | > 0.98 |
| PKS0637-75 | 98.94 | −75.27 | 6.7 | 230 ± 10 | 10 | 12.63 ± 0.56 | < 0.01 | 230 ± 10 | 34 | 11.94 ± 2.26 | < 0.01 | 0.69 ± 2.33 |
| PKS0637-75 | 98.94 | −75.27 | 6.7 | 393 ± 8 | 14 | 12.91 ± 0.26 | < 0.01 | — | — | < 12.04 | — | > 0.87 |
| PKS0637-75 | 98.94 | −75.27 | 6.7 | 348 ± 5 | 19 ± 9 | 13.40 ± 0.20 | < 0.01 | 348 ± 5 | 44 ± 25 | 12.76 ± 0.26 | < 0.01 | 0.64 ± 0.33 |
| PKS0637-75 | 98.94 | −75.27 | 6.7 | 290 ± 9 | 37 ± 19 | 13.52 ± 0.17 | < 0.01 | 290 ± 9 | 43 | 12.60 ± 0.68 | 0.03 ± 0.04 | 0.92 ± 0.70 |
| IRAS Z06229-6434 | 98.94 | −64.61 | 6.7 | 448 ± 2 | 27 ± 4 | 13.53 ± 0.05 | < 0.01 | 448 ± 2 | 16 ± 3 | 12.79 ± 0.10 | < 0.01 | 0.74 ± 0.11 |
| IRAS Z06229-6434 | 98.94 | −64.61 | 6.7 | 366 ± 10 | 22 | 13.22 ± 0.33 | < 0.01 | 366 ± 10 | 23 | 12.05 ± 0.91 | 0.02 ± 0.04 | 1.17 ± 0.96 |
| IRAS Z06229-6434 | 95.78 | −64.61 | 6.7 | 306 ± 8 | 35 ± 11 | 13.33 ± 0.13 | < 0.01 | 306 ± 8 | 30 | 12.51 ± 0.45 | 0.04 ± 0.05 | 0.82 ± 0.47 |
| IRAS Z06229-6434 | 95.78 | −64.61 | 6.7 | — | — | < 12.43 | — | 262 ± 10 | 10 | 12.02 ± 0.67 | 0.02 ± 0.03 | < 0.41 |
| IRAS Z06229-6434 | 95.78 | −64.61 | 6.7 | 394 ± 7 | 11 | 12.86 ± 0.66 | < 0.01 | 394 ± 7 | 36 | 12.20 ± 0.71 | < 0.01 | 0.66 ± 0.97 |
| UVQSJ045415.95-611626.6 | 73.57 | −64.61 | 7.8 | 319 ± 5 | 19 ± 11 | 13.50 ± 0.57 | < 0.01 | 319 ± 5 | 32 ± 6 | 13.49 ± 0.17 | < 0.01 | 0.02 ± 0.59 |
| UVQSJ045415.95-611626.6 | 73.57 | −61.27 | 7.8 | 394 ± 10 | 28 ± 14 | 13.12 ± 0.19 | < 0.01 | — | — | < 12.35 | — | > 0.77 |
| UVQSJ045415.95-611626.6 | 73.57 | −61.27 | 7.8 | 216 ± 5 | 15 ± 8 | 12.84 ± 0.15 | < 0.01 | — | — | < 12.39 | — | > 0.46 |
| UVQSJ045415.95-611626.6 | 73.57 | −61.27 | 7.8 | 302 ± 13 | 38 ± 6 | 13.88 ± 0.26 | < 0.01 | 302 ± 13 | 40 ± 22 | 12.96 ± 0.55 | 0.02 ± 0.02 | 0.92 ± 0.61 |
| RBS563 | 69.62 | −61.80 | 8.0 | 311 ± 13 | 24 | 13.90 ± 0.59 | < 0.01 | 311 ± 13 | 30 ± 16 | 13.39 ± 0.54 | < 0.01 | 0.51 ± 0.80 |
| RBS563 | 69.62 | −61.80 | 8.0 | 353 ± 26 | 39 ± 18 | 14.09 ± 0.38 | 0.05 ± 0.04 | 353 ± 26 | 38 ± 16 | 13.46 ± 0.42 | 0.12 ± 0.12 | 0.63 ± 0.56 |
| RBS563 | 69.62 | −61.80 | 8.0 | 157 ± 37 | 40 | 13.21 ± 0.53 | < 0.01 | — | — | < 12.28 | — | > 0.93 |
| RBS563 | 46.90 | −72.83 | 8.0 | 253 ± 5 | 20 ± 9 | 13.70 ± 0.14 | < 0.01 | 253 ± 5 | 27 ± 6 | 13.30 ± 0.12 | < 0.01 | 0.40 ± 0.18 |
| ESO031-G08 | 46.90 | −72.83 | 9.7 | 303 ± 12 | 24 ± 9 | 13.78 ± 0.27 | < 0.01 | 303 ± 12 | 26 ± 11 | 13.27 ± 0.44 | < 0.01 | 0.51 ± 0.52 |
| ESO031-G08 | 46.90 | −72.83 | 9.7 | 262 ± 11 | 19 | 13.64 ± 0.36 | < 0.01 | 262 ± 11 | 20 | 12.95 ± 0.74 | < 0.01 | 0.69 ± 0.82 |
| ESO031-G08 | 40.79 | −72.28 | 9.7 | 214 ± 18 | 30 | 13.41 ± 0.38 | < 0.01 | 214 ± 18 | 27 | 13.13 ± 0.70 | 0.03 ± 0.05 | 0.28 ± 0.80 |
| UKS0242-724 | 40.79 | −72.28 | 11.3 | 194 ± 12 | 24 | 13.21 ± 1.01 | 7.52 ± 17.44 | 194 ± 12 | 50 ± 8 | 13.04 ± 0.18 | 5.31 ± 2.16 | 0.18 ± 1.02 |
| UKS0242-724 | 40.79 | −72.28 | 11.3 | 253 ± 11 | 50 | 13.67 ± 0.74 | 0.74 ± 1.27 | 253 ± 11 | 31 ± 17 | 13.10 ± 0.28 | 1.54 ± 0.99 | 0.57 ± 0.79 |
| UKS0242-724 | 66.50 | −57.20 | 11.3 | 312 ± 17 | 36 ± 12 | 13.41 ± 0.76 | < 0.01 | 312 ± 17 | 31 ± 19 | 12.86 ± 0.33 | < 0.01 | 0.55 ± 0.83 |
| 1H0419-577 | 66.50 | −57.20 | 12.1 | 311 ± 1 | 24 ± 2 | 13.55 ± 0.03 | < 0.01 | 311 ± 1 | 14 ± 3 | 12.73 ± 0.06 | 0.03 ± 0.00 | 0.82 ± 0.06 |
| 1H0419-577 | 65.22 | −56.85 | 12.1 | 356 ± 3 | 20 ± 0 | 13.09 ± 0.09 | 0.02 ± 0.00 | 356 ± 3 | 29 ± 7 | 12.65 ± 0.08 | 0.05 ± 0.01 | 0.43 ± 0.12 |
| HE0419-5657 | 65.22 | −56.85 | 12.6 | 300 ± 4 | 25 ± 4 | 13.62 ± 0.10 | < 0.01 | 300 ± 4 | 32 ± 11 | 12.78 ± 0.12 | 0.03 ± 0.01 | 0.84 ± 0.15 |
| HE0419-5657 | 30.56 | −76.33 | 12.6 | 360 ± 10 | 35 ± 7 | 13.50 ± 0.15 | < 0.01 | — | — | < 12.36 | — | > 1.14 |
| PKS0202-76 | 30.56 | −76.33 | 13.3 | 258 ± 6 | 20 ± 8 | 13.34 ± 0.14 | < 0.01 | 258 ± 6 | 29 ± 16 | 12.82 ± 0.23 | < 0.01 | 0.52 ± 0.27 |
| PKS0202-76 | 30.56 | −76.33 | 13.3 | 294 ± 7 | 16 | 12.80 ± 0.48 | < 0.01 | — | — | < 12.45 | — | > 0.35 |
| RBS563 | 69.91 | −53.19 | 14.9 | — | — | < 12.68 | — | — | — | < 12.28 | — | — |
| HE0435-5304 | 69.21 | −52.98 | 15.1 | 296 ± 16 | 22 | 13.31 ± 0.45 | < 0.01 | — | — | < 12.29 | — | > 1.01 |
| HE0435-5304 | 69.21 | −52.98 | 15.1 | 350 ± 24 | 33 | 13.34 ± 0.39 | < 0.01 | 350 ± 24 | 24 | 12.75 ± 0.74 | < 0.01 | 0.59 ± 0.83 |
| HE0435-5304 | 69.21 | −52.98 | 15.1 | 250 ± 5 | 25 | 13.18 ± 0.32 | < 0.01 | 250 ± 5 | 9 | 12.67 ± 0.65 | < 0.01 | 0.51 ± 0.72 |
| HE0439-5254 | 70.05 | −52.80 | 15.2 | 303 ± 2 | 34 ± 6 | 13.40 ± 0.06 | < 0.01 | 303 ± 2 | 15 ± 6 | 13.10 ± 0.08 | < 0.01 | 0.30 ± 0.10 |
| HE0439-5254 | 70.05 | −52.80 | 15.2 | 243 ± 4 | 15 | 12.92 ± 0.16 | < 0.01 | — | — | < 12.28 | — | > 0.64 |
| PKS0558-504 | 89.95 | −50.45 | 16.9 | — | — | < 12.81 | — | — | — | < 12.20 | — | — |
| PKS0355-483 | 59.34 | −48.20 | 20.3 | 203 ± 7 | 44 ± 6 | 13.24 ± 0.10 | < 0.01 | — | — | < 12.17 | — | > 1.07 |
| FAIRALL9 | 20.94 | −58.81 | 22.6 | 185 ± 3 | 13 ± 8 | 12.87 ± 0.29 | 5.51 ± 3.66 | 185 ± 3 | 24 ± 5 | 12.79 ± 0.06 | 3.95 ± 0.57 | 0.08 ± 0.29 |
| RBS1992 | 350.46 | −70.45 | 23.4 | 244 ± 8 | 33 ± 11 | 13.27 ± 0.13 | 0.02 ± 0.00 | 236 ± 4 | 12 | 12.31 ± 0.14 | 0.09 ± 0.03 | 0.96 ± 0.19 |
| RBS1992 | 350.46 | −70.45 | 23.4 | 173 ± 8 | 32 ± 11 | 13.27 ± 0.14 | < 0.01 | 186 ± 3 | 20 ± 5 | 12.72 ± 0.06 | 0.05 ± 0.01 | 0.55 ± 0.15 |
| HE0331-4112 | 53.28 | −41.03 | 26.6 | — | — | < 12.50 | — | — | — | < 12.21 | — | — |
| RBS144 | 15.11 | −51.23 | 28.6 | 181 ± 10 | 44 ± 7 | 13.11 ± 0.14 | < 0.01 | — | — | < 12.19 | — | > 0.92 |
| HE0246-4101 | 42.03 | −40.81 | 28.8 | — | — | < 12.98 | — | 179 ± 6 | 25 ± 10 | 12.91 ± 0.09 | < 0.01 | < 0.08 |
| HE0153-4520 | 28.81 | −45.10 | 29.0 | — | — | < 12.38 | — | 197 ± 5 | 12 | 12.22 ± 0.14 | 0.02 ± 0.01 | < 0.16 |
| HE0226-4110 | 37.06 | −40.95 | 29.8 | 203 ± 3 | 12 | 12.70 ± 0.36 | 0.02 ± 0.01 | 203 ± 3 | 14 ± 5 | 12.58 ± 0.07 | 0.03 ± 0.01 | 0.12 ± 0.36 |
| HE0226-4110 | 37.06 | −40.95 | 29.8 | 160 ± 6 | 33 ± 14 | 13.20 ± 0.15 | < 0.01 | 160 ± 6 | 11 | 12.09 ± 0.21 | 0.03 ± 0.02 | 1.11 ± 0.26 |
| HE0038-5114 | 37.06 | −40.95 | 30.0 | — | — | < 12.83 | — | — | — | < 12.65 | — | — |
| HE2336-5540 | 354.81 | −55.40 | 30.9 | — | — | < 12.71 | — | — | — | < 12.37 | — | — |
| HE0003-5023 | 1.43 | −50.12 | 30.9 | — | — | < 12.54 | — | — | — | < 12.07 | — | — |
| IRAS F21325-6237 | 324.09 | −62.40 | 32.5 | 170 ± 1 | 14 ± 2 | 13.32 ± 0.04 | < 0.01 | 170 ± 1 | 9 ± 6 | 12.17 ± 0.13 | 0.02 ± 0.01 | 1.15 ± 0.14 |
| HE2305-5315 | 347.16 | −52.98 | 33.9 | — | — | < 12.50 | — | — | — | < 12.97 | — | — |

Best-fit model parameters for C IV and Si IV for components associated with Magellanic CGM with $v_{LSR} > 150$ km s$^{-1}$. For each sightline, the LMC impact parameter $\rho$, centroid velocity $v$, Doppler parameter (linewidth; $b$), column density $\log_{10}(N/cm^{-2})$ and photoionized fraction $f_{PI}$ ($f_{PI} = N_{Cloudy}/N_{Obs}$) for both ions are given. Components with $f_{PI} \lesssim 0.1$ are considered as not photoionized and shown in Fig. 3. Linewidths from data with low S/N ratio spectra or that are fixed in the fitting process do not show errors. Uncertainties in this table correspond to 1σ standard deviations. Further fit results for all ions and sightlines can be viewed at https://github.com/Deech08/HST_MagellanicCorona.

**Extended Data Table 2 | Voigt profile model parameters for O VI**

| Source Name | RA [deg] | Dec [deg] | $\rho_{\mathrm{LMC}}$ [kpc] | $v_{\mathrm{O\,VI}}$ [km s$^{-1}$] | $b_{\mathrm{O\,VI}}$ [km s$^{-1}$] | $\log_{10}\left(\dfrac{N_{\mathrm{O\,VI}}}{\mathrm{cm}^2}\right)$ |
|---|---|---|---|---|---|---|
| IRAS Z06229-6434 | 95.78 | -64.61 | 6.7 | $231 \pm 12$ | 40 | $14.07 \pm 0.14$ |
| IRAS Z06229-6434 | 95.78 | -64.61 | 6.7 | $352 \pm 9$ | 40 | $14.30 \pm 0.11$ |
| IRAS Z06229-6434 | 95.78 | -64.61 | 6.7 | $466 \pm 11$ | $36 \pm 16$ | $14.07 \pm 0.16$ |
| ESO031-G08 | 46.90 | -72.83 | 9.7 | $231 \pm 10$ | 35 | $14.17 \pm 0.11$ |
| ESO031-G08 | 46.90 | -72.83 | 9.7 | $308 \pm 5$ | 25 | $14.33 \pm 0.10$ |
| 1H0419-577 | 66.50 | -57.20 | 12.1 | $302 \pm 7$ | $16 \pm 8$ | $14.03 \pm 0.13$ |
| 1H0419-577 | 66.50 | -57.20 | 12.1 | $358 \pm 11$ | $28 \pm 18$ | $13.90 \pm 0.22$ |
| RBS144 | 15.11 | -51.23 | 28.6 | $199 \pm 9$ | 32 | $13.88 \pm 0.12$ |
| HE0226-4110 | 37.06 | -40.95 | 29.8 | $180 \pm 8$ | 26 | $13.72 \pm 0.13$ |
| IRAS F21325-6237 | 324.09 | -62.40 | 32.5 | $165 \pm 6$ | $31 \pm 10$ | $14.04 \pm 0.10$ |

Best-fit model parameters for O VI for components associated with Magellanic CGM with $v_{\mathrm{LSR}} > 150\,\mathrm{km\,s^{-1}}$. Linewidths without errors are shown when they are fixed in the fitting process. Further fit results for all ions and sightlines can be viewed at https://github.com/Deech08/HST_MagellanicCorona.

**Extended Data Table 3 | Partial Spearman rank-order correlation tests**

| Ion | # of Detections | $r_{N\,\rho_{\mathrm{LMC}};\,|B_{\mathrm{MS}}|}$ | $p$-value | $r_{N\,|B_{\mathrm{MS}}|;\,\rho_{\mathrm{LMC}}}$ | $p$-value |
|---|---|---|---|---|---|
| C IV | 17 | −0.793 | <0.001 | 0.323 | **0.222** |
| Si IV | 17 | −0.524 | 0.037 | 0.224 | **0.362** |
| Si III | 22 | −0.664 | 0.001 | −0.158 | **0.494** |
| Si II | 21 | −0.583 | 0.007 | −0.328 | **0.159** |
| C II | 20 | −0.525 | 0.021 | −0.076 | **0.758** |
| Al II | 18 | −0.534 | 0.027 | −0.468 | **0.058** |
| Fe II | 12 | −0.143 | **0.676** | −0.561 | **0.072** |
| O I | 13 | −0.117 | **0.717** | −0.693 | 0.012 |

Tests of the relation between ion column density and LMC impact parameter, after removing the effects of absolute Magellanic Stream latitude ($r_{N\rho_{\mathrm{LMC};}|B_{\mathrm{MS}}|}$) and between ion column density and absolute Magellanic Stream latitude, after removing the effects of the LMC impact parameter ($r_{N|B_{\mathrm{MS}}|;\rho_{\mathrm{LMC}}}$). $P$-values for each partial test are shown in boldface if they are greater than the 0.05 threshold, not allowing the null hypothesis of no correlation to be rejected.