## [Peer Review File · Nature]

Manuscript Title: Observations of a Magellanic Corona

Reviewer Comments & Author Rebuttals

Reviewer Reports on the Initial Version:

Referees' comments:

Referee #1 (Remarks to the Author):

This manuscript details the first observational evidence for an extended gaseous halo (known as the circumgalactic medium) around the Large Magellanic Cloud, a massive dwarf galaxy near the Milky Way. This evidence confirms the predictions made by a previous publication in *Nature* (Lucchini et al. 2020), who used hydrodynamic simulations to show that the inclusion of warm-hot ionized gas (a “Magellanic Corona”) around the LMC leads to the natural reproduction of the Magellanic Stream, and its leading arm, whose existence have been previously difficult to explain. Lucchini et al. predict the existence of hot, highly ionized gas in the vicinity of the LMC. This manuscript includes direct evidence of such gas. As such, the topic of the paper is not novel, however the context of the discovery and quality of data certainly are.

The observational methods implemented in this manuscript are state-of-the-art, and comparable to other published works detailing similar discoveries. Based on a cursory review of the literature, this observation includes an impressively detailed measurement of a CGM by way of number of sightlines. The figures included present this data in a manner that supports their analysis and adds clarity to the interpretation.

The statistical techniques included in this manuscript are standard for the field and relevant to the analysis of the authors. Further, the tests support the validity of their data and the claims made within the text. The conclusions made in this manuscript are supported by reasonable interpretations of the data, particularly through the comparison of different ionization models to constrain the temperature and ionization level of the observed gas. Such methods are standard for the field and when applied appropriately, generate robust predictions.

In terms of scientific content that could improve this paper, it would be interesting to explore a differentiation between gas that is associated to the LMC alone versus gas that could potentially be traced to the SMC in order to make predictions about the gas content of satellite galaxies in Magellanic systems. This suggestion goes beyond the intended analysis of the paper, and serves to highlight its potential impact. If the data is available, such an analysis could improve the paper, however this suggestion is not necessary to warrant publication. The included measurements are sufficient to provide powerful constraints to analytical & numerical models, as well as inform further observational science on the LMC itself.

The abstract, while somewhat technical for *Nature*, provides a satisfactory summary of the measurements and conclusions made in the manuscript. Perhaps the introduction should include a

few sentences justifying the interest in CGM from a galaxy evolution perspective, tying the results to a wider variety of scientific applications.

Referee #2 (Remarks to the Author):

The manuscript presents observational evidence for a $T \sim 10^{5.5}$ K corona for the Large Magellanic Cloud (LMC) based on absorption line studies of low (e.g., SiII, SiIII, CII) and high (SiIV, CIV, and OVI) ions toward a sample of QSO sightlines within impact parameters of ~ 50 kpc from the LMC. The manuscript shows declining column density radial profiles of ions from the LMC. Furthermore, the manuscript includes several ionization modeling analyses, which show that low and high ions near the LMC are not likely to co-exist in the same phases.

The effort of detecting the LMC's corona and the evidence shown in this manuscript would be of great interest to many researchers in the CGM and broader communities. It is not straightforward to find conclusive evidence for the LMC's corona given the complicated environment the galaxy is in (e.g., the Milky Way's CGM, the overall Magellanic System, the SMC, etc.). Many of my comments and questions below are related to how the sample is designed, and how cleanly one can isolate the LMC's corona absorption signals from other potential contamination sources.

Line 38-42:

The manuscript uses 28 sightlines within impact parameters of 50 kpc, which is half the virial radius of the LMC. I was wondering why only sightlines within 50 kpc were selected? As shown in Extended Figure 7 as well as many works referenced in the manuscript, there are a lot more UV sightlines at 50-100 kpc from the LMC and near the Magellanic Stream, the Leading Arm, as well as toward high Magellanic latitude regions. Given the richness of these additional data and their proximity to the LMC within its virial radius, it would be informative and necessary to analyze these data together with what is presented in the manuscript. It is possible that at impact parameters further than 50 kpc, the influence from the Milky Way's CGM might dominate the absorption signals. If so, a different trend in $\log N$ in Figure 2 beyond 50 kpc might be expected. It would be useful if the manuscript provides further discussion on the choice of sightline selection and considers including sightlines at 50-100 kpc in their analyses, given that the contamination from the Milky Way's CGM is likely to be significant.

Additionally, the manuscript does not adequately describe how the sample of 28 sightlines were selected. Was it based on some signal-to-noise ratio cut from a broader archival search of available sightlines in MAST?

A couple sightlines at small impact parameters (e.g., RXJ0503.1-6634 at 3.2 kpc and PKS0552-640 at 5.6 kpc) seem to go through the disk of the LMC based on Figure 1. If this were the case, I was wondering whether the ions detected toward these sightlines are instead probing winds or outflows from the LMC, as is the case toward CAL-F in Barger et al. (2016)? Furthermore, these ions could be in the corona of the LMC, which means that the distance to these ions may be less clear because they are observed in a down-the-barrel viewing angle. If so, how does this geometry effect affect the radial profile trend in Figure 2 and in Extended Data Figure 2?

Line 88-91:

“We find that the component line widths of SiIV tend to be broader than those of SiII, as expected if the high-ions exist in a hotter phase than the low-ions, although the CIV and CII line widths are harder to distinguish, which suggests that non-thermal broadening dominates for both the low-ion and high-ion gas”. It is not clear to me what message this sentence is meant to deliver. The first half argues that temperature plays a role in the line width difference between SiIV and SiII, while the second half instead suggests that non-thermal broadening is the key. Overall, the analyses seem to suggest that the pair of SiII-SiIV and that of CII-CIV provide different interpretation on the thermal dynamic states of the multi-phase gas. This seems to contradict the manuscript’s previous conclusion that low-ions are in the same phase (line 71-73) from Cloudy modeling, while CIV and SiIV co-exist in another hotter phase (line76-77). It would be useful if the discussion in this paragraph can be restructured to provide a more clear analysis on the component kinematics of low and high ions.

Line 92-97:

“In particular the standard deviation ... This may suggest that the triply-ionized and singly-ionized atoms are tracing approximately co-spatial gas, while OVI may trace physically distinct gas”. As noted in the Methods, the spectral resolution for COS is 15-20 km/s (Line 288-289) and for FUSE is 20 km/s (Line 296). These values are very similar to the standard derivations derived for the centroid velocity differences between the low and high ions. I was wondering how spectral resolutions may affect the evaluation of line centroid differences here since this would be critical for understanding whether the low and high ions are co-spatial.

Line 325-326:

“If the component structure of the low and high ions match, they are flagged after the fitting process...”. I was wondering what criteria were used to determine whether the low and high ions’ component structures are matched? This is important because results from component fits are later used to compare the line widths and centroid velocities between the low and high ions. If “match” means the velocity centroids are close to each other in low and high ions, then by design one should expect a small centroid velocity differences in the low and high ion measurements. I was wondering how does this analysis design impact the interpretation of the high and low ions’ kinematic similarities?

Line 151-156:

“Several sightlines in our sample Gas associated with the Stream...”. I appreciate the effort of describing possible contamination and previous work here. However, the discussion does not seem to be as adequate as the rest of the manuscript. For example, how would the logN radial profile in Figure 2 change if absorption affected by MW Seyfert Flare (Bland-Hawthorn+2019) or the Magellanic Stream (especially those “on-stream” sightlines in Fox et al. 2014) were removed from the sample?

Line 154-156 and Line 477-478:

The manuscript suggests that the radial profile in low and high ions would be hard to explain if the ions were associated with the Magellanic Stream. And the correlation between $\log N(\text{CIV})$ and ρ_{LMC} is slightly stronger ($\tau=-0.54$) than that between $\log N(\text{CIV})$ and absolute Magellanic latitude ($\tau=-0.49$). The two τ values from the Kendall's Tau rank correlation coefficient tests do not seem to be that different from each other. The test results do not seem to be conclusive enough to suggest that an LMC corona scenario is more likely than an extended Magellanic envelope scenario (as proposed by Fox et al. 2014). I was wondering if the authors can comment on the significance of the Kendall's Tau rank correlation coefficient tests here.

Furthermore, are the same tests being done for other ions (e.g., OVI, SiIV, SiIII, SiII)? Do they also show slightly stronger correlation between $\log N(\text{ion})$ and ρ_{LMC} as well?

Line 156-159:

"This radial profile is similar to that seen in a previous survey of the CGM of 43 low mass, $z < 0.1$ dwarf galaxies..." The manuscript has not adequately described what comparison has been done to support this statement. I was wondering if the authors could include a figure showing the radial profile from this manuscript and those from Bordoloi et al. (2014) as well as Lehner et al. (2020)?

Figure 3:

In Figure 3, it seems that the brown symbols in the middle panel should be plotted as squares instead of circles if they were to represent the non-equilibrium case?

Extended Data Table 2:

There seems to be missing negative signs in the Dec column.

Author Rebuttals to Initial Comments:

1 May, 2022

Dear Dr. Sage,

We thank you and the two anonymous referees for the detailed comments and suggestions in the last version of our manuscript. We submit to your attention the revised version of our paper and our point-by-point responses below.

Referee 1:

This manuscript details the first observational evidence for an extended gaseous halo (known as the circumgalactic medium) around the Large Magellanic Cloud, a massive dwarf galaxy near the Milky Way. This evidence confirms the predictions made by a previous publication in *Nature* (Lucchini et al. 2020), who used hydrodynamic simulations to show that the inclusion of warm-hot ionized gas (a “Magellanic Corona”) around the LMC leads to the natural reproduction of the Magellanic Stream, and its leading arm, whose existence have been previously difficult to explain. Lucchini et al. predict the existence of hot, highly ionized gas in the vicinity of the LMC. This manuscript includes direct evidence of such gas. As such, the topic of the paper is not novel, however the context of the discovery and quality of data certainly are.

The observational methods implemented in this manuscript are state-of-the-art, and comparable to other published works detailing similar discoveries. Based on a cursory review of the literature, this observation includes an impressively detailed measurement of a CGM by way of number of sightlines. The figures included present this data in a manner that supports their analysis and adds clarity to the interpretation.

The statistical techniques included in this manuscript are standard for the field and relevant to the analysis of the authors. Further, the tests support the validity of their data and the claims made within the text. The conclusions made in this manuscript are supported by reasonable interpretations of the data, particularly through the comparison of different ionization models to constrain the temperature and ionization level of the observed gas. Such methods are standard for the field and when applied appropriately, generate robust predictions.

In terms of scientific content that could improve this paper, it would be interesting to explore a differentiation between gas that is associated to the LMC alone versus gas that could potentially be traced to the SMC in order to make predictions about the gas content of satellite galaxies in Magellanic systems. This suggestion goes beyond the intended analysis of the paper, and serves to highlight its potential impact. If the data is available, such an analysis could improve the paper, however this suggestion is not necessary to warrant publication. The included measurements are sufficient to provide powerful constraints to analytical & numerical models, as well as inform further observational science on the LMC itself.

Authors:

We agree that separating LMC gas from SMC gas would indeed be scientifically interesting and we thank the referee for pointing this out. 6 of our 28 sightlines lie closer on the sky to the SMC than the LMC (and 3 of these 6 probe the Magellanic Bridge). However, proximity alone is not enough to associate an absorber to either galaxy. Previous work that examined tidally-stripped gas in the Magellanic Stream used gas-phase metallicity to make such a distinction. Unfortunately, a similar method would not be effective when considering the hotter phases of gas in interfaces or the corona, as this gas is expected to have a lower metallicity, and be of a more primordial origin, distinct from the ISM of

the present-day LMC or SMC. Additionally, given that the LMC has a mass 10 times greater than that of the SMC (Besla et al. 2012), the SMC is not expected to have its own corona. Instead a single corona from the LMC will dominate the Magellanic System and envelope both galaxies. This is the case in current simulations (eg. Lucchini et al. 2020, 2021), in which the SMC does not contain a significant amount of gas to populate a CGM. We have briefly described this in the current manuscript lines 101 - 103.

Referee 1:

The abstract, while somewhat technical for Nature, provides a satisfactory summary of the measurements and conclusions made in the manuscript. Perhaps the introduction should include a few sentences justifying the interest in CGM from a galaxy evolution perspective, tying the results to a wider variety of scientific applications.

We have simplified the technical points of the summary paragraph and introduction to better fit with the editorial comments and requirements. This has resulted in a new summary paragraph that combines the introductory text from the submitted version lines 3-37, into a single and more concise summary paragraph spanning the current draft from lines 3-20.

Referee 2:

The manuscript presents observational evidence for a $T \sim 10^{5.5}$ K corona for the Large Magellanic Cloud (LMC) based on absorption line studies of low (e.g., SiII, SiIII, CII) and high (SiIV, CIV, and OVI) ions toward a sample of QSO sightlines within impact parameters of ~ 50 kpc from the LMC. The manuscript shows declining column density radial profiles of ions from the LMC. Furthermore, the manuscript includes several ionization modeling analyses, which show that low and high ions near the LMC are not likely to co-exist in the same phases.

The effort of detecting the LMC's corona and the evidence shown in this manuscript would be of great interest to many researchers in the CGM and broader communities. It is not straightforward to find conclusive evidence for the LMC's corona given the complicated environment the galaxy is in (e.g., the Milky Way's CGM, the overall Magellanic System, the SMC, etc.). Many of my comments and questions below are related to how the sample is designed, and how cleanly one can isolate the LMC's corona absorption signals from other potential contamination sources.

Line 38-42:

The manuscript uses 28 sightlines within impact parameters of 50 kpc, which is half the virial radius of the LMC. I was wondering why only sightlines within 50 kpc were selected? As shown in Extended Figure 7 as well as many works referenced in the manuscript, there are a lot more UV sightlines at 50-100 kpc from the LMC and near the Magellanic Stream, the Leading Arm, as well as toward high Magellanic latitude regions. Given the richness of these additional data and their proximity to the LMC within its virial radius, it would be informative and necessary to analyze these data together with what is presented in the manuscript. It is possible that at impact parameters further than 50 kpc, the influence from the Milky Way's CGM might dominate the absorption signals. If so, a different trend in $\log N$ in Figure 2 beyond 50 kpc might be expected. It would be useful if the manuscript provides further discussion on the choice of sightline selection and considers including sightlines at 50-100 kpc in their analyses, given that the contamination from the Milky Way's CGM is likely to be significant.

Additionally, the manuscript does not adequately describe how the sample of 28 sightlines were selected. Was it based on some signal-to-noise ratio cut from a broader archival search of available sightlines in MAST?

Authors:

We understand and appreciate these comments, and we agree that a complete investigation of a Magellanic Corona would in principle include additional sightlines that extend much further in impact parameters from the LMC, as would typically be done in more distant extragalactic studies. The basic problem is that the LMC is so close to the Milky Way that going out beyond ~ 50 kpc requires sampling sightlines more than one radian on the sky from the LMC. At these large angular separations, it is very difficult to reliably associate high-velocity absorption components with the LMC, because the absorption overlaps spatially and kinematically with multiple contaminating sources including the Magellanic Stream, the Leading Arm, and other MW high velocity clouds. This is a fundamental difference from CGM experiments on external galaxies at higher redshift (COS Halos, COS Dwarfs, etc) where even large impact parameters correspond to small angular sizes on the sky, and where the small-angle approximation is sufficient in calculating impact parameters, and more complex 3D distributions of gas are negligible.

Our current sample (28 sightlines) was designed to consider all archival sightlines with COS G130M and G160M observations with $S/N > 7$ and within an angular separation of 45 degrees. At higher angular separations, our ability to realistically calculate an impact parameter is greatly diminished and would require a true 3D model to physically locate the absorbers. In order to expand our sample to larger angular separations, and thus sample a larger extent of the virial radius of the LMC, the analysis and results would become very dependent on the adopted 3D model. Since we are only 50 kpc away from the LMC, gas at angular separations greater than 45 degrees has an implied impact parameter that is larger than the distance of the gas to our position in the Milky Way. At larger and larger distances, it becomes more likely to attribute high velocity gas to the Milky Way and its CGM, rather than the Magellanic System.

Additionally, the Magellanic Corona is expected to behave differently than the C IV halos of isolated galaxies, because it continues to undergo major disruptions as the LMC approaches the Milky Way (e.g. Besla+, Pardy+, Lucchini+). We have already seen a drop-off in the observed covering fraction of C IV absorbers at large impact parameters greater than 25 kpc to just 30%, possibly a result of the Corona being tidally truncated from the interactions. This drop in covering fraction also means that our mass estimates converge quickly, with differences of under 0.1 dex in the total mass of the high ions when integrated to 35 kpc vs. the mass when extrapolated and integrated out to 100 kpc.

All that being said, to further explore the referee's suggestion, we looked into 25 additional archival sightlines located at angular separations of the LMC between 45 and 90 degrees. We have performed the same data reduction process and measured the C IV absorption using an apparent optical depth method (Savage & Sembach 1991). This analysis resulted in only 9 detections of C IV, or a covering fraction in this region of 36%, with the remaining 16 sightlines showing non-detections. Of these 9 detections, 5 sightlines are in the proximity of the Leading Arm, and so may not probe the diffuse Corona. The 4 directions away from the Leading Arm are still qualitatively consistent with the declining radial profile we observe. Below, we share a map, in the same style as Figure 1, panel a), that includes these 25 additional sightlines of C IV measurements, with non-detections shown as open, white circles. However, we note that these 25 additional directions have not undergone photoionization modeling, which would require measurements of several low and high-ion absorption features in every sightline, and so would be labor intensive. This photoionization modeling process may explain some of the 9 sightlines showing C IV absorption, and further reduce our covering fraction when only considering collisionally ionized gas.

To summarize, it is nearly impossible to distinguish LMC gas from the Milky Way at large separation angles without a 3D model. The addition of sightlines at larger angular separations will not likely impact our radial profile, or sample a significant mass of the Magellanic CGM, and would result in the addition of a high number of non-detections. We therefore think that such an extension of our sample would be difficult to interpret and would lead to inconclusive outcomes. Therefore, after careful consideration we respectfully ask to keep our sample as is, prioritizing quality over quantity. To better explain our sample design, we have added a brief section to the Methods section, subtitled “*Impact Parameters and Projection Effects*,” which highlights the complexity of assigning impact parameters at the large angular separations involved. Our updated manuscript lines 21-24 also more clearly describes the angular separation cut-off and S/N criteria used to select our archival sample.

Related to this point, we found that our assumed geometry to calculate impact parameters needed to be refined. In our previously submitted manuscript, we assumed the gas existed in a flat plane set at the distance of the LMC, with (impact parameter) = $\tan(\text{angular separation}) \times (\text{distance to LMC})$. This assumption does not provide the closest possible radial distance from the LMC, which is found using (impact parameter) = $\sin(\text{angular separation}) \times (\text{distance to LMC})$, so this is the equation used in the revised manuscript. At small angular separations, there is very little difference between the two methods, but at the large angular separations present in our work, using the revised equation reduces the physical extent of our sample to sightlines within an impact parameter of 35 kpc, spanning approximately one-third of the virial radius estimates of the LMC of 100-130 kpc. Because the LMC halo is already within the virial radius of the MW, it is expected to be tidally truncated, hence smaller values for its virial

radius are likely in the present day. We note that this change has not impacted any of our results, and we still see a statistically significant radial profile, and can calculate masses that converge to amounts that are within 0.2 dex of our previously quoted amounts, comparable to their estimated uncertainties.

Referee 2:

A couple sightlines at small impact parameters (e.g., RXJ0503.1-6634 at 3.2 kpc and PKS0552-640 at 5.6 kpc) seem to go through the disk of the LMC based on Figure 1. If this were the case, I was wondering whether the ions detected toward these sightlines are instead probing winds or outflows from the LMC, as is the case toward CAL-F in Barger et al. (2016)? Furthermore, these ions could be in the corona of the LMC, which means that the distance to these ions may be less clear because they are observed in a down-the-barrel viewing angle. If so, how does this geometry effect affect the radial profile trend in Figure 2 and in Extended Data Figure 2?

Authors:

This is an interesting point. The inner-most sightlines do indeed probe a complex region, where winds or outflows, the disk of the LMC itself, and projection effects complicate the observations. CAL-F is actually in our sample, but listed under the name RX_J0503.1-6634 (this is the lowest impact parameter sightline in our sample). We agree that the inner-most sightlines may be tracing winds or outflows, and we have noticed an apparent deficit of high-ions in these sightlines, which is discussed on PDF page 4 and explored in extended data figure 6. This high-ion deficit could indeed be caused by ionization or mechanical removal of the gas by winds, just as the referee suggested, or alternatively it could be caused by the projection effect/geometry. If these observed absorbers were located at a larger distance from the LMC of around 15-25 kpc (still along the line of sight but at a different distance from the Sun), then they would fit it well with the radial profile we see. However, the low-ion columns do not show a similar deficit, which argues against the projection-effect explanation, and instead suggests the high-ion effect is related to winds or ionization in the inner-CGM region.

Referee 2:

Line 88-91:

“We find that the component line widths of SiIV tend to be broader than those of SiII, as expected if the high-ions exist in a hotter phase than the low-ions, although the CIV and CII line widths are harder to distinguish, which suggests that non-thermal broadening dominates for both the low-ion and high-ion gas”. It is not clear to me what message this sentence is meant to deliver. The first half argues that temperature plays a role in the line width difference between SiIV and SiII, while the second half instead suggests that non-thermal broadening is the key. Overall, the analyses seem to suggest that the pair of SiII-SiIV and that of CII-CIV provide different interpretation on the thermal dynamic states of the multi-phase gas. This seems to contradict the manuscript’s previous conclusion that low-ions are in the same phase (line 71-73) from Cloudy modeling, while CIV and SiIV co-exist in another hotter phase (line 76-77). It would be useful if the discussion in this paragraph can be restructured to provide a more clear analysis on the component kinematics of low and high ions.

Authors:

We apologize for the confusion with our explanation of results related to the component kinematics. The line widths of both low ions and high ions are broad in our measurements, but given the spectral resolution of our COS spectra

(15-20 km/s), we are unable to determine whether a finer velocity structure is hidden within our measurements. The line widths we find are typically broader than would be expected from thermal broadening, which ultimately makes it hard to use the line widths to diagnose gas temperatures directly. To simplify our arguments, we have removed lines 87-97 from our original manuscript submission and overall are de-emphasizing the kinematic results. What we can say is that the line widths, while not conclusive, are still consistent with the multiphase picture for the Magellanic CGM. We still do see a significant difference in the population of linewidths we have measured between singly and triply ionized C and Si (which we show in Extended Data Figure 4, panels c and d). The most notable difference in kinematics is seen when considering the velocity centroids of O VI absorption with that of the lower ionization states. When considering the nearest neighboring velocity components, the OVI velocity centroids have large offsets from the other ions, which suggest that they are tracing a physically distinct medium. In our updated manuscript, lines 41-43 mention this kinematic result and lines 80-86 describe them in the context of our conclusions. Our Methods section lines 470-502 provide a summary of our kinematics results and uncertainties involved.

Referee 2:

Line 92-97:

“In particular the standard deviation ... This may suggest that the triply-ionized and singly-ionized atoms are tracing approximately co-spatial gas, while OVI may trace physically distinct gas”. As noted in the Methods, the spectral resolution for COS is 15-20 km/s (Line 288-289) and for FUSE is 20 km/s (Line 296). These values are very similar to the standard derivations derived for the centroid velocity differences between the low and high ions. I was wondering how spectral resolutions may affect the evaluation of line centroid differences here since this would be critical for understanding whether the low and high ions are co-spatial.

Authors:

We agree that the low spectral resolution impacts our ability to fully draw conclusions from our kinematic results. The precise uncertainties of centroid velocities are difficult to model, as they depend on the S/N of the spectra, but our uncertainties in the centroid velocities found during our voigt-profile fitting process should reflect these uncertainties (in the range of 5-12 km/s, see Extended Data Tables 1 and 2). These errors are propagated through our measurements of velocity centroid differences as shown in lines 485-486. The difference between the standard deviations of the velocity centroid differences between OVI and other ions is 7 (+12/-8) km/s. While a difference of 0 is still within 1 standard deviation, it is less likely, and our interpretation of OVI tracing a distinct medium is still consistent with this possibility, but not conclusive when only considering this test.

Referee 2:

Line 325-326:

“If the component structure of the low and high ions match, they are flagged after the fitting process...”. I was wondering what criteria were used to determine whether the low and high ions’ component structures are matched? This is important because results from component fits are later used to compare the line widths and centroid velocities between the low and high ions. If “match” means the velocity centroids are close to each other in low and high ions, then by design one should expect a small centroid velocity differences in the low and high ion measurements. I was wondering how does this analysis design impact the interpretation of the high and low ions’ kinematic similarities?

Authors:

We agree that this component matching process is inherently biased towards small centroid velocity differences and difficult to account for. As the referee suspects, this matching process is indeed done through looking for velocity centroids that are close to each other. However, we note that when considering the low-ions in comparison with C IV and Si IV, the velocity structure seen in absorption across a broad range of velocities tended to match in the sense that close to the same number of absorption components were seen and fit for the low and high ions in sightlines where high ions were detected. Since ultimately, this bias is difficult to quantify and overcome, we believe our decision to de-emphasize the kinematic results as explained above is the most appropriate action. We have added a note on this inherent bias to our Methods section lines 472-476.

Referee 2:

Line 151-156:

“Several sightlines in our sample ... Gas associated with the Stream...”. I appreciate the effort of describing possible contamination and previous work here. However, the discussion does not seem to be as adequate as the rest of the manuscript. For example, how would the logN radial profile in Figure 2 change if absorption affected by MW Seyfert Flare (Bland-Hawthorn+2019) or the Magellanic Stream (especially those “on-stream” sightlines in Fox et al. 2014) were removed from the sample?

Authors:

The sightlines in our current sample do not extend into the region of the Magellanic Stream that is affected by the MW Seyfert flare ionization cone in the Bland-Hawthorn+ (2019) models, so we do not expect our log N profile to change. This was another reason why a larger, more extended sample was avoided when we designed our sample. This information has been added into the manuscript in lines 91-92.

As for the Magellanic Stream, the separation from the Corona is not straightforward, especially when considering the simulation predictions from Lucchini et al. (2020) that find that ~50% of the Stream’s mass originated in the Magellanic Corona, so the two structures are closely connected. We verified that if we remove the “On-stream” sightlines from Fox et al. 2014 from our sample, we still see a declining radial profile based on the sightlines extended off the direction of the Stream (towards the right in Figure 1). Therefore we conclude that our radial profile is not dominated by the influence of the Magellanic Stream.

Referee 2:

Line 154-156 and Line 477-478:

The manuscript suggests that the radial profile in low and high ions would be hard to explain if the ions were associated with the Magellanic Stream. And the correlation between logN(CIV) and rho_LMC is slightly stronger (tau=-0.54) than that between logN(CIV) and absolute Magellanic latitude (tau=-0.49). The two tau values from the Kendall’s Tau rank correlation coefficient tests do not seem to be that different from each other. The test results do not seem to be conclusive enough to suggest that an LMC corona scenario is more likely than an extended Magellanic envelope scenario (as proposed by Fox et al. 2014). I was wondering if the authors can comment on the significance of the Kendall’s Tau rank correlation coefficient tests here.

Furthermore, are the same tests being done for other ions (e.g., OVI, SiIV, SiIII, SiII)? Do they also show slightly stronger correlation between logN(ion) and rho_LMC as well?

Authors:

We agree that our statistical test used to compare the correlation between our observed $\log N$ and ρ_{LMC} and absolute Magellanic latitude are largely inconclusive, and the slight differences are not significant enough to make a conclusion. To improve our analysis here, we instead consider a partial correlation test using the Spearman rank-order test. This test calculates a correlation coefficient between two variables, after removing the effects of a third variable. For our situation, we consider the partial correlation between $\log N$ and ρ_{LMC} , after removing the effects of absolute Magellanic latitude, as well as the partial correlation between $\log N$ and absolute Magellanic latitude, after removing the effects of ρ_{LMC} . This is done for several ions (C IV, Si IV, Si II, C II, Al II, Fe II, O I) and finds a statistically significant result for a stronger correlation seen between $\log N$ and ρ_{LMC} for most ions (C IV, Si IV, Si II, C II, Al II), while Fe II is inconclusive and O I shows a significantly stronger correlation between $\log N$ and absolute Magellanic latitude. Given that O I often traces cooler gas, we conclude that this is consistent with our interpretation, with the Magellanic CGM centered on the LMC dominating the effects we observe, but O I may be more strongly tracing tidally stripped, cooler gas in the Magellanic Stream. This analysis is explained in detail in the Methods in lines 522-533, with the results of the partial correlation statistical tests shown in Extended Data Table 3.

Referee 2:

Line 156-159:

“This radial profile is similar to that seen in a previous survey of the CGM of 43 low mass, $z < 0.1$ dwarf galaxies...”. The manuscript has not adequately described what comparison has been done to support this statement. I was wondering if the authors could include a figure showing the radial profile from this manuscript and those from Bordoloi et al. (2014) as well as Lehner et al. (2020)?

Authors:

We have included an additional panel (b) to Extended Data Figure 2 that shows CIV column density measurements from Bordoloi et al. (2014) and Lehner et al. (2020) along with those from our work around the LMC, with the x-axis showing the impact parameters normalized by virial radius. We note that this figure may be difficult to make direct comparisons with since the range of uncertainty in the virial radius estimates of the LMC (100-130 kpc) result in a relatively large difference in the the LMC profile when normalized by virial radius. To best show this, we display our LMC observations using a R_{vir} value of 115 ± 15 kpc, with error bars corresponding to this uncertainty. In addition, the estimates of the virial radius (or R_{200}) from Bordoloi et al. (2014) and Lehner et al. (2020) are said to have an uncertainty of 50%, which further increases our difficulty in making a realistic comparison of profiles. This is described in the Methods lines 375-383 and mentioned in the main text lines 97-100.

Referee 2:

Figure 3:

In Figure 3, it seems that the brown symbols in the middle panel should be plotted as squares instead of circles if they were to represent the non-equilibrium case?

Extended Data Table 2:

There seems to be missing negative signs in the Dec column.

Authors:

We apologize for these two mistakes in our previous manuscript. We have fixed the plotting symbols in Figure 3 and corrected the misprinted Dec values in Extended Data Table 2.

Summary of Other Changes to Manuscript:

Based on editorial instructions, we have significantly shortened the main text of our manuscript. We have also added figure captions and extended data captions in the text, after the references, now lines 175-282.

We have removed repetitive statements and shortened the introduction to a more concise form (previously lines 3-42; now lines 3-20).

We have removed most of the content from previous lines 53-66, which is described in more detail in the Methods section (now lines 360 - 366).

We have shortened the details of previous lines 67 - 86, which is described in more detail in the Methods section (now lines 387-444). Key results from these lines have been stated more concisely now in lines 34-43.

We have removed some previously stated predictions about observing the photoionized 10^4 K phase in optical emission, to keep our results and discussion more focused on the $10^{5.5}$ K Corona phase in this paper (previously lines 162-166)

We have updated our impact parameter values for all sources as described above, which includes updates to the values shown in Extended Data Tables 1 and 2, as well as the scaling for the x-axis in Figure 2, Figure 3, and Extended Data Figure 2. Contours of the Impact parameter shown in Figure 1 and Extended Data Figure 7 have also been updated accordingly. Statistical tests on the radial profile have been recomputed using the correct impact parameter values, with minimal changes to results. Mass estimates (shown in lines 62-73 of the current manuscript and in Figure 3) have been updated using the correct impact parameter values, with differences generally within previously quoted uncertainties.

We have modified our statistical tests to be consistent in the p-value significance threshold we have adopted throughout the paper (p-value = 0.05), which we now mention upfront in the Methods line 469.

The typeset formatting of the article has been changed as requested in editorial comments, such that all Figure legends and Extended Data legends are displayed after the main text references. To improve clarity while reading for review purposes, these legends are also displayed below each display item at this time.

Sincerely,

Dhanesh Krishnarao, Andrew J. Fox, Elena D'Onghia, Bart P. Wakker, Frances H. Cashman, J. Christopher Howk, Scott Lucchini, David M. French, Nicolas Lehner

Reviewer Reports on the First Revision:

Referees' comments:

Referee #1 (Remarks to the Author):

The authors have resubmitted a revised manuscript as well as a response to the comments given by both referees. In response to my original comments, they have updated the summary paragraph and introduction. The updated summary paragraph includes a more concise description of the work done by the authors and satisfactorily address my initial concerns of over-technicality.

Further, the authors responded to my comments on the distinction between LMC-associated gas and SMC-associated gas. They have provided additional details on the type of analysis this would include, and indicate that the SMC is not expected to have its own corona. I would clarify that dwarf galaxies of similar mass to the SMC are indeed expected to host observable quantities of gas in their ISM and/or CGM in isolation, but that this gas can be removed or mixed with the ambient gas content via interactions with their host. Ultimately, the same conclusion is reached that the SMC at present day is not expected to host a distinguishable gas component, so the relevance to the observational techniques employed by the authors is somewhat tangential. I leave it up to the authors whether they wish to include a discussion on this point.

Referee #2 (Remarks to the Author):

I have reviewed the updated manuscript and the rebuttal letter submitted by the authors. I thank the authors for their hard work at addressing my comments. The updated arguments made by the authors are well-supported with sufficient analyses and demonstration. I do not have further major comments on the updated manuscript.

One minor comment: the corrected maximum impact parameter (35 kpc instead of 50 kpc) seems to have been misquoted in a few places: Page 4/Line 59, Extended Data Figure 1, Page 31/Equation 1, and Page 32/Line 572.

Author Rebuttals to First Revision:

Referee 1:

The authors have resubmitted a revised manuscript as well as a response to the comments given by both referees. In response to my original comments, they have updated the summary paragraph and introduction. The updated summary paragraph includes a more concise description of the work done by the authors and satisfactorily address my initial concerns of over-technicality.

Further, the authors responded to my comments on the distinction between LMC-associated gas and SMC-associated gas. They have provided additional details on the type of analysis this would include, and indicate that the SMC is not expected to have its own corona. I would clarify that dwarf galaxies of similar mass to the SMC are indeed expected to host observable quantities of gas in their ISM and/or CGM in isolation, but that this gas can be removed or mixed with the ambient gas content via interactions with their host. Ultimately, the same conclusion is reached that the SMC at present day is not expected to host a distinguishable gas component, so the relevance to the observational techniques employed by the authors is somewhat tangential. I leave it up to the authors whether they wish to include a discussion on this point.

Authors:

We have added a brief paragraph to better discuss the SMC and why we do not expect to see its own corona when in proximity to the LMC and MW.

Referee 2:

I have reviewed the updated manuscript and the rebuttal letter submitted by the authors. I thank the authors for their hard work at addressing my comments. The updated arguments made by the authors are well-supported with sufficient analyses and demonstration. I do not have further major comments on the updated manuscript.

One minor comment: the corrected maximum impact parameter (35 kpc instead of 50 kpc) seems to have been misquoted in a few places: Page 4/Line 59, Extended Data Figure 1, Page 31/Equation 1, and Page 32/Line 572.

Authors:

We have updated the misquoted sections of the paper and extended data to properly display the correct impact parameter values.

We would like to thank you again for your time and consideration in reviewing our manuscript for publication.

Sincerely,

Dhanesh Krishnarao, Andrew J. Fox, Elena D'Onghia, Bart P. Wakker, Frances H. Cashman, J. Christopher Howk, Scott Lucchini, David M. French, Nicolas Lehner